# Calibration of a simple and a complex model of global marine biogeochemistry

Iris Kriest[1]

[1]GEOMAR Helmholtz-Zentrum für Ozeanforschung Kiel, Düsternbrooker Weg 20, D-24105 Kiel, Germany

*Correspondence to:* Iris Kriest (ikriest@geomar.de)

**Abstract.**

The assessment of the ocean biota's role in climate change is often carried out with global biogeochemical ocean models that contain many components, and involve a high level of parametric uncertainty. Because many data that relate to tracers included in a model are only sparsely observed, assessment of model skill is often restricted to tracers that can be easily measured and

assembled. Examination of the models' fit to climatologies of inorganic tracers, after the models have been spun up to steady state, is a common, but computationally expensive procedure to assess model performance and reliability. Using new tools that have become available for global model assessment and calibration in steady state, this paper examines two different model types - a complex seven-component model (MOPS), and a very simple four-component model (RetroMOPS) - for their fit to dissolved quantities. Before comparing the models, a subset of their biogeochemical parameters has been optimised against

annual mean nutrients and oxygen. Both model types fit the observations almost equally well. The simple model contains only two nutrients, oxygen and dissolved organic phosphorus (DOP). Its misfit and large scale tracer distributions are sensitive to the parameterisation of DOP production and decay. The spatio-temporal decoupling of nitrogen and oxygen, and processes involved in their uptake and release, renders oxygen and nitrate valuable tracers for model calibration. In addition, the non-conservative nature of these tracers (with respect to their upper boundary condition) introduces the global bias (fixed nitrogen

and oxygen inventory) as a useful additional constraint on model parameters. Dissolved organic phosphorus at the surface behaves antagonistically to phosphate, and suggests that observations of this tracer - although difficult to measure - may be an important asset for model calibration.

## 1    Introduction

Global biogeochemical ocean models are now routinely used to assess the ocean biota's role in climate change. Although

these models have become ever more complex with respect to the number of biogeochemical tracers they contain, they are often calibrated only against a subset of their components, mostly nutrients, oxygen, and components of the carbon cycle (e.g. Bacastow and Maier-Reimer, 1991; Ilyina et al., 2013; Cocco et al., 2013; Cabre et al., 2015).

There has been an intensive discussion about the necessary level of marine ecosystem model complexity, mostly on a theoretical basis, or in a local or regional context (e.g., Anderson, 2005; Le Quere, 2006; Flynn, 2006; Anderson, 2006; Leles

et al., 2016; Shimoda and Arhonditsis, 2016). It remains an open question whether additional complexity is of advantage for

representing biogeochemical processes and tracers on a global scale (i.e., for processes acting on rather long time and large space scales). For example, Kriest et al. (2010, 2012) found no large differences when comparing model skill with respect to oxygen and phosphate across a range of models of different complexity, but quite large effects of parameter settings when applying a coarse examination of the parameter space.

However, a thorough and dense scan of the parameter space would be required for a fair assessment of the virtues of models of different complexity. Such a scan usually requires many model evaluations, which, given long equilibration time scales of coupled global models (Khatiwala, 2008; Wunsch and Heimbach, 2008; Primeau and Deleersnijder, 2009; Siberlin and Wunsch, 2011), is difficult to carry out. For assessment of only surface properties and processes a short model spin-up may be sufficient; however, on a global scale, many centuries to millennia of coupled model simulations are necessary, in order to remove the drift in biogeochemical tracer fields and fit to observed properties (Kriest and Oschlies, 2015; Seferian et al., 2016).

Only recently tools have become available that allow for a reduction in simulation times, such as the Transport Matrix Method (TMM, Khatiwala et al., 2005), which replaces the ocean circulation model with an efficient "offline" circulation, or methods that solve for steady state tracer fields using Newton's method. The latter require either the inversion of the Jacobian (e.g., Kwon and Primeau, 2006), or apply matrix-free Newton-Krylov (MFNK, Khatiwala, 2008; Li and Primeau, 2008) to compute the steady-state solution. Surrogate-based optimisation replaces the original and computationally expensive model by a so-called surrogate, which is created from a less accurate but computationally cheaper model. The latter is corrected to reduce the misalignment between the two solutions. Priess et al. (2013) applied this method, together with the TMM to recover parameters of a simple global biogeochemical model; the surrogate in their case consisted of shorter (decades) spin-ups.

The gain in computational efficiency resulting from these methods can then be used for a systematic calibration of global biogeochemical models. For example, Kwon and Primeau (2006, 2008) used global climatological data sets of phosphate, inorganic carbon, and alkalinity to calibrate a simple global biogeochemical model. The misfit between observed and simulated phosphate was used by DeVries et al. (2014) to calibrate parameters related to particle properties in a simple two-component, nutrient-restoring model. In a similar approach Holzer et al. (2014) optimised parameters for opal production and dissolution against observed silicate. Letscher et al. (2015) switched between a complex and a simple model of ocean biogeochemistry to estimate production and decay rates of dissolved organic phosphorus on a global scale.

All these biogeochemical models employed in global parameter estimates were of a low level of biogeochemical complexity. One reason for this restriction might be associated with the variety of time scales associated with more complex models. Piwonski and Slawig (2016) used MFNK to evaluate the steady state of simple and complex biogeochemical models. They noted that "[...] for more complex models the Newton method requires more attention to solver parameter settings [...]" (Piwonski and Slawig, 2016), which may be related to the highly nonlinear structure of these models. The nonlinearity, and the large number of parameters, also complicates their simultaneous optimisation (Ward et al., 2010). On a global scale, these problems are amplified by the sparsity of observations of organism groups, particularly of higher trophic levels. Observations of dissolved inorganic constituents, on the other hand, are much more frequent, and therefore provide more information on the spatio-temporal variability of these tracers.

Recently, Kriest et al. (2017) combined the TMM with an estimation of distribution algorithm (Covariance Matrix Adaption Evolution Strategy, CMA-ES), to optimise six biogeochemical parameters of a seven component model against global climatologies of annual mean phosphate, nitrate and oxygen. They showed that annual mean tracer concentrations did not provide much information on parameters related to the dynamic biological processes taking place in the euphotic zone, but that parameters related to long time and large space scales (e.g., the remineralisation length scale or so-called "Martin $b$"; see also Kriest et al., 2012) could be estimated from these observations. The large uncertainty associated with surface parameter estimates can be attributed to the relatively small volume of the surface ocean, leading to a misfit that is dominated by deep ocean observations.

Replacing the misfit function by a metric that targets at the surface ocean, and/or contains additional observations that provide information on plankton could be one way to resolve this indeterminacy. Alternatively, one could skip these parameters from optimisation and focus on parameters more tightly connected to the meso- and bathypelagic ocean. A more drastic measure is lies in downscaling the biogeochemical model to a simpler system, that only contains components with a counterpart in global, quasi-synoptic data sets. The latter procedure may help to elucidate which level of complexity is required to represent and investigate global distribution and patterns of biogeochemical tracers.

This paper examines the the latter two potential solutions: Firstly, I investigate if parameters related to oxidant-dependent decay in the mesopelagial are better constrained by this type of misfit function. This is done by replacing four parameters of the optimisation carried out by Kriest et al. (2017) by parameters related to oxidant-affinity of remineralisation, and - to account for the possible alterations in fixed nitrogen turnover - by the maximum nitrogen fixation rate. Secondly, given the successful parameter optimisation of simpler models noted above, and also to acknowledge the fact, that these models have been popular and quite successful in global simulations of ocean biogeochemistry (e.g. Bacastow and Maier-Reimer, 1990, 1991; Matear and Hirst, 2003; Kwon and Primeau, 2006; Dutkiewicz et al., 2006), this paper presents an optimised model, which has been derived from downscaling the seven-component model MOPS (Kriest and Oschlies, 2015; Kriest et al., 2017) to a model that retains only three abiotic dissolved tracers (phosphate, nitrate, and oxygen) and one biotic tracer (dissolved organic phosphorus; DOP). This new model, which I refer to as "RetroMOPS", includes the oxidant-dependency of MOPS, but is otherwise very similar to models applied earlier in global models. In contrast to some of these models (Marchal et al., 1998; Najjar et al., 2007) it assumes no relaxation to observed tracer fields, but simulates fully prognostic changes in surface production, as in Bacastow and Maier-Reimer (1991); Maier-Reimer (1993); Matear and Hirst (2003); Parekh et al. (2005).

After a brief presentation of model MOPS (Kriest and Oschlies, 2015), the downscaled model RetroMOPS is introduced, followed by an outline on circulation, optimisation and experimental design (section 2). In section 3 results from optimisation of both MOPS and RetroMOPS are presented and discussed. The paper closes with conclusions drawn from these experiments.

## 2 Models, experiments, and optimisations

### 2.1 Model MOPS

Model MOPS (Kriest and Oschlies, 2015) is based on phosphorus, and simulates seven compartments. Phosphate, phytoplankton, zooplankton, dissolved organic phosphorus (DOP) and detritus are calculated in units of mmol P m$^{-3}$. Oxygen is coupled to the P-cycle with a constant stoichiometry given by $R_{-\mathrm{O2:P}}$. Aerobic remineralisation of organic matter follows a saturation curve, with half-saturation constant $K_{\mathrm{O2}}$. Aerobic remineralisation ceases when oxygen declines; at the same time, denitrification takes over, as long as nitrate is available above a defined threshold, $DIN_{\mathrm{min}}$. Like the oxic process, suboxic remineralisation follows a saturation curve for oxidant nitrate, with half-saturation constant $K_{\mathrm{DIN}}$. MOPS does not explicitly resolve the different oxidation states of inorganic nitrogen (nitrite, N$_2$O, ammonium), but assumes immediate coupling of the different processes involved in nitrate reduction, the end-product being dinitrogen (see also Paulmier et al., 2009; Kriest and Oschlies, 2015). All organic components are characterised by a constant N:P stoichiometry of $d = 16$. Loss of fixed nitrogen is balanced by a simple parameterisation of nitrogen fixation by cyanobacteria, which relaxes the nitrate-to-phosphate ratio to $d$ with a time constant, $\mu_{\mathrm{NFix}}^{*}$. In the long term nitrogen fixation balances the simulated loss of fixed nitrogen via denitrification, although they may occur in distant areas (see Kriest and Oschlies, 2015, for more details).

Detritus sinks with a vertically increasing sinking speed: $w = a\,z$. Assuming a constant degradation rate $r$, in equilibrium this would result in a particle flux curve given by $F(z) \propto z^{-b}$, with $b = r/a$. For better comparison with values of $b$ derived from observations (e.g. Martin et al., 1987; Van Mooy et al., 2002; Buesseler et al., 2007), and with the simpler model RetroMOPS (see below), $a$ is expressed in terms of $b$ (assuming constant, nominal $r = 0.05$ d$^{-1}$). A fraction of detritus deposited at the sea floor (at the bottom of the deepest vertical box) is buried instantaneously in some hypothetical sediment. The fraction buried depends on the deposition rate onto the sediment. Non-buried detritus is resuspended into the last box of the water column, where it is treated as regular detritus. The phosphorus budget is closed on an annual time scale through resupply via river runoff. More details about the biogeochemical model and parameters, and their effects on model behaviour can be found in Kriest and Oschlies (2013) and Kriest and Oschlies (2015).

### 2.2 Model RetroMOPS

MOPS' structure has been simplified by skipping the explicit simulation of phytoplankton, zooplankton, and detritus (see Fig. S1). The remaining equations of, and functional relationships between, phosphate, nitrate, oxygen and DOP have been parameterised similar to MOPS. Because the downscaled model resembles so many features of earlier biogeochemical models simulated in a global context (e.g., Bacastow and Maier-Reimer, 1991; Maier-Reimer, 1993; Matear and Hirst, 2003), but keeps the oxidant dependency of MOPS, the model is named "RetroMOPS".

### 2.2.1 Primary production

Like MOPS, RetroMOPS calculates primary production only in the euphotic zone, which, in the current configuration, is confined to the upper two numerical layers ($k_{EZ} = 2$, $z = 0 - 120$ m). As in Kriest et al. (2010) and Kriest et al. (2012) phytoplankton is parameterised with a constant concentration of $\overline{PHY} = 0.02$ mmol P m$^{-3}$, which is the mean phytoplankton concentration in the upper 120 m of two optimised model setups MOPS$^{oS}$ and MOPS$^{oD}$ (see below). Using this constant phytoplankton concentration RetroMOPS calculates light- and nutrient dependent primary production $P$ in each layer $k$ as:

$$P(k) = \begin{cases} \mu_{PHY} \, \overline{PHY} \min\left( f(I(k)), \frac{L(k)}{K_{PHY} + L(k)} \right) & : \quad k \leq k_{EZ} \\ 0 & : \quad k > k_{EZ} \end{cases} \tag{1}$$

where $f(I(k))$ defines light-limitation, $\mu_{PHY}$ is the temperature-dependent maximum growth rate of phytoplankton, and $L$ determines the limiting nutrient: $L(k) = \min(PO_4(k), DIN(k)/d)$ (see Kriest and Oschlies, 2015, for more details). Note that, with the given parameters for nutrient and light affinity, the resulting specific growth rates ($P(k)/PHY$) of optimised MOPS and RetroMOPS are quite similar (0.127 d$^{-1}$ for MOPS and 0.102 d$^{-1}$ for RetroMOPS).

### 2.2.2 The fate of primary production: Export, DOP production and remineralisation

Instead of resolving heterotrophic processes (zooplankton grazing, excretion and egestion) at the sea surface explicitly, in RetroMOPS a fraction $\sigma$ of organic matter fixed photosynthetically is immediately released as dissolved organic phosphorus, DOP. DOP then decays to phosphate and nitrate with a constant rate $\lambda$. To allow for a potential, fast recycling loop at the surface, RetroMOPS parameterises an additional decay rate, $\lambda_s$, that affects DOP only in the first two layers. By doing so, the model mimics multiple DOP fractions with different decay rates, as observed by Hopkinson et al. (2002). The remaining fraction of production, $1 - \sigma$, of each layer in the euphotic zone is exported to the layers below, where it immediately remineralises to nutrients, following a power-law of depth. The discretised form for flux $F$ into box $j$ from all (surface) source layers $k$, with $1 \leq k \leq k_{EZ}$ is then given by:

$$F(j) = \sum_{k=1}^{k=k_{EZ}} P(k)(1-\sigma)\Delta z(k) \left( \frac{z(j)}{z(k+1)} \right)^{-b} \qquad \text{for} \quad j > k \,, \tag{2}$$

where $\Delta z(k)$ denotes the thickness of a numerical (source) layer, and $z(j)$ is the depth of the upper boundary of layer $j$. The flux divergence, $D = dF/dz$, for any box $j$ in discretised form is defined by

$$D(j) = \frac{F(j-1) - F(j)}{\Delta z(j)} \tag{3}$$

Neglecting oxidant dependency of decay, the entire flux divergence $D(j)$ would be released as phosphate and nitrate, with equivalent oxidant consumption. It is, however, possible that oxidants become depleted at some location. Earlier models in

this case continued the degradation of organic matter, thereby implicitly assuming unspecified oxidants (e.g., Marchal et al., 1998; Matear and Hirst, 2003; Najjar et al., 2007; Kriest et al., 2010, 2012). In contrast, RetroMOPS, like MOPS, accounts for suppression of remineralisation (oxic/suboxic) in the absence of sufficient oxidants, by assuming saturation curves for the limitation by either oxygen or nitrate. The amount of organic matter available for oxidation is given by the decay of dissolved

organic matter, $\lambda DOP$, and by the flux divergence, $D(j)$ (Eqn. 3). The discretised flux divergence, that can actually be remineralised with available oxidants (oxygen and/or nitrate), $D^{\text{eff}}(j)$, is then determined by

$$D^{\text{eff}}(j) = D(j) \left( s_{\text{O2}}(j) + s_{\text{DIN}}(j) \right) \qquad , \tag{4}$$

where $s_{\text{O2}}(j)$ and $s_{\text{DIN}}(j)$ represent the oxidant limitation terms, expressed as saturation curves $l_{\text{O2}}$ and $l_{\text{NO3}}$ for either oxygen (oxic remineralisation) or nitrate (denitrification), with half-saturation constants $K_{\text{O2}}$ and $K_{\text{DIN}}$. Denitrification is

further inhibited by oxygen via $(1 - l_{\text{O2}})$, resulting in $s_{\text{O2}}(j) + s_{\text{DIN}}(j) = 1$; (see also Equations 15-27 of Kriest and Oschlies, 2015). In all models oxic remineralisation only takes place down to a a lower threshold of $O_2 = 4 \text{mmol m}^{-3}$. The lower threshold for denitrification is determined by parameter $DIN_{\text{min}}$, and subject to optimisation.

The flux divergence that cannot be remineralised under the given concentrations of oxidants is added as additional flux divergence to the layer below:

$$D(j+1) \quad = \quad D(j+1) + (D(j) - D^{\text{eff}}(j)) \frac{\Delta z(j)}{\Delta z(j+1)} \tag{5}$$

where again $D^{\text{eff}}(j+1)$ is evaluated. In the bottom layer the remaining flux that has not been remineralised in the water column eventually enters the sediment.

### 2.2.3 Benthic exchanges

Models that implicitly assume unspecified oxidants often prescribe a zero boundary flux, i.e. all organic matter in the last

bottom box is degraded instantaneously (e.g., Marchal et al., 1998; Matear and Hirst, 2003; Najjar et al., 2007; Yool et al., 2011). Both MOPS and RetroMOPS have to take "leftover" organic matter flux into account, that arrives undegraded at the sea floor because of incomplete remineralisation in the water column. The explicit detritus compartment in MOPS allows for only partial burial at the sea floor, which may result in detritus accumulation in the deepest model box (see Kriest and Oschlies, 2013). Because there is no such detritus compartment in RetroMOPS, all flux arriving at the sea floor is buried immediately.

Therefore, MOPS and RetroMOPS differ with respect to their lower boundary condition.

### 2.2.4 Nitrogen fixation

Both RetroMOPS and MOPS do not explicitly simulate cyanobacteria, but assume zero net growth of these organisms, parameterised as an immediate release of fixed nitrogen as nitrate:

$$
S(k) = \begin{cases} \mu^*_{\text{NFix}}\, f_1(\text{T}(k))\, f_2(\text{DIN}(k), \text{PO}_4(k)) & : \quad k \leq k_{\text{EZ}} \\ 0 & : \quad k > k_{\text{EZ}} \end{cases} \tag{6}
$$

$f_1$ parameterises the temperature dependence of nitrogen fixation with a second order polynomial approximation of the function by Breitbarth et al. (2007). $f_2$ regulates the relaxation of the nitrate:phosphate ratio towards the global observed stoichiometric ratio of $d = 16$. The maximum nitrogen fixation $\mu^*_{\text{NFix}}$ (mmol N m$^{-3}$d$^{-1}$) parameterises an implicit cyanobacteria population. As in MOPS, on long time scales nitrogen fixation balances the simulated loss of fixed nitrogen via denitrification, although the regions of nitrogen loss and gain can be spatially segregated (Kriest and Oschlies, 2015).

### 2.2.5 Source-minus-sinks

Combining the above mentioned processes and interactions, the time rate of change in each layer $k$ for phosphate, nitrate, oxygen, and DOP due to biogeochemical processes are

$$
S^{\text{PO4}} = \underbrace{-P}_{\text{production}} + \underbrace{\lambda_{\text{s}}\text{DOP}}_{\text{surface decay}} + \underbrace{[D + \lambda\text{DOP}]\,[s_{\text{O2}} + s_{\text{DIN}}]}_{\text{decay and flux divergence}} \tag{7}
$$

$$
S^{\text{DOP}} = \underbrace{\sigma_{DOP}\,P}_{\text{release}} - \underbrace{\lambda_{\text{s}}\text{DOP}}_{\text{surface decay}} - \underbrace{\lambda\text{DOP}\,[s_{\text{O2}} + s_{\text{DIN}}]}_{\text{decay}} \tag{8}
$$

$$
S^{\text{O2}} = \underbrace{R_{-\text{O2:P}}\,P}_{\text{production}} - \underbrace{R_{-\text{O2:P}}\,\lambda_{\text{s}}\text{DOP}}_{\text{surface decay}} - \underbrace{R_{-\text{O2:P}}\,[D + \lambda\text{DOP}^*]\,s_{\text{O2}}}_{\text{decay and flux divergence}} \tag{9}
$$

$$
S^{\text{DIN}} = \underbrace{-d\,P}_{\text{production}} + \underbrace{S}_{\text{N-fixation}} + \underbrace{d\,\lambda_{\text{s}}\text{DOP}}_{\text{surface decay}} + \underbrace{[D + \lambda\text{DOP}]\,[s_{\text{O2}}\,d - s_{\text{DIN}}\,R_{-\text{DIN:P}}]}_{\text{decay and flux divergence}} \tag{10}
$$

Summarising, RetroMOPS is similar to model "N-DOP" of Kriest et al. (2010, 2012), to the phosphorus component of the model presented by Parekh et al. (2005), or to the models presented by Bacastow and Maier-Reimer (1991) and Maier-Reimer (1993), the exception being details of primary production at the sea surface, and the explicit parameterisation of oxidant-dependent remineralisation. By assuming constant cyanobacteria biomass, and a relaxation of the nitrate:phosphate ratio via immediate release of fixed nitrogen, its parameterisation of nitrogen fixation is similar to the one described by Maier-Reimer et al. (2005) and Ilyina et al. (2013). Because RetroMOPS lacks explicit phytoplankton, zooplankton and detritus it has eight tunable parameters less than MOPS.

## 2.3 Circulation and physical transport

All model simulations apply the Transport Matrix Method (TMM; Khatiwala, 2007, `github.com/samarkhatiwala/tmm`) for tracer transport, with monthly mean transport matrices (TMs), wind, temperature and salinity (for air-sea gas exchange) derived from a 2.8° global configuration of the MIT ocean model, with 15 levels in the vertical, as described in Marshall

et al. (1997) and Dutkiewicz et al. (2005). The circulation model was forced with climatological annual cycles of wind, heat and freshwater fluxes, and subject to a weak restoring of surface temperature and salinity to observations. Its configuration is similar to that applied in the Ocean Carbon-Cycle Model Intercomparison Studies (OCMIP) (Orr et al., 2000), which has been assessed against observations of temperature, salinity and mixed layer depth (Doney et al., 2004), CFCs (Dutay et al., 2002; Matsumoto et al., 2004) and radiocarbon (Matsumoto et al., 2004; Graven et al., 2012). Overall, its performance is comparable

to other global models.

Using this efficient offline approach, a time step length of 1/2 day for tracer transport and 1/16 day for biogeochemical interactions, simulation of 3000 years requires about 0.5-1.5 hrs on 4 nodes (24 core Intel Xeon Ivybridge) at a High Performance Computing Centre (`www.hlrn.de`). After 3000 years most tracers have approached steady state (see also Kriest and Oschlies, 2015, for long time trends of MOPS simulated in a different circulation), and the transient of the misfit function

becomes very small (see Fig. S2). The last year is used for model analysis and evaluation of the misfit function.

## 2.4 Optimisation algorithm

Optimisation of parameters is carried out using an Estimation of Distribution Algorithm, namely the Covariance Matrix Adaption Evolution Strategy (CMA-ES; Hansen and Ostermeier, 2001; Hansen, 2006). The application of this algorithm to the coupled biogeochemistry-TMM framework has shown good performance with respect to quality and efficiency (in terms of

function evaluations), and is described only briefly below. More details about the algorithm, its setup and coupling to the global biogeochemical model can be found in Kriest et al. (2017).

Let $n$ be the number of biogeochemical parameters to be estimated. In each iteration ("generation") the algorithm defines a population of $\lambda$ individuals (biogeochemical parameter vectors of length $n$), with $\lambda = 10$ (derived from the default parameter $\lambda = 4 + 3\ln(n)$, Hansen and Ostermeier, 2001). The candidate vectors are sampled from a multi-variate normal-distribution,

which generalises the usual normal distribution, also known as Gaussian distribution, from $\mathbb{R}$ to the vector space $\mathbb{R}^n$.

Following the simulation of these $\lambda$ individual model setups to steady state (3000 years), the misfit function is evaluated, and information of the current, as well as previous generations is used to update the probability distribution in $\mathbb{R}^n$ such that the likelihood to sample good solutions increases. Usually, the realisation of the probability distribution update ensures that information of former solutions fades out slowly, resisting for several iterations. Therefore, the population (the number of

model simulations per generation) in CMA-ES is smaller, and of less computational demand, than in classical evolutionary algorithms. Nevertheless, CMA-ES can still, to a certain degree, perform well with misfit functions characterised by a rough topography (Kriest et al., 2017).

## 2.5 Misfit function

As in Kriest et al. (2017) the misfit to observations $J$ is defined as the root-mean-square error RMSE between simulated and observed annual mean phosphate, nitrate, and oxygen concentrations (Garcia et al., 2006a, b), mapped onto the three-dimensional model geometry. Although regridding the observations onto the coarser model geometry removes some of the variability, this method is computationally more efficient in an optimisation framework. Also, a sensitivity study with a similar coupled model showed that accounting for the variance inherent in the observational data, and arising from regridding did not have a large influence on the misfit (Kriest et al., 2010).

Deviations between model and observations are weighted by the volume of each individual grid box, $V_{\mathrm{i}}$, expressed as fraction of total ocean volume, $V_{\mathrm{T}}$. The resulting sum of weighted deviations is then normalised by the global mean concentration of the respective observed tracer:

$$J = \sum_{j=1}^{3} J(j) = \sum_{j=1}^{3} \frac{1}{\overline{o_j}} \sqrt{\sum_{i=1}^{N} (m_{i,j} - o_{i,j})^2 \frac{V_i}{V_{\mathrm{T}}}} \tag{11}$$

$j = 1, 2, 3$ indicates the tracer type and $i = 1, ..., N$ are the model locations for $N = 52749$ model grid boxes. $\overline{o_j}$ is the global average observed concentration of the respective tracer. $m_{i,j}$ and $o_{i,j}$ are model and observations, respectively. By weighting each individual misfit with volume, $J$ serves more as a long time-scale geochemical estimator, in contrast to a misfit function that e.g. focuses on (rather fast) turnover in the surface layer, or resolves the seasonal cycle.

## 2.6 Optimisation of MOPS

Based on a "hand-tuned", a priori setup of MOPS (Kriest and Oschlies, 2015), which hereafter is referred to as MOPS$^{\mathrm{r}}$, Kriest et al. (2017) presented an optimisation of mostly surface-related parameters (hereafter referred to as MOPS$^{\mathrm{oS}}$). They chose a very wide range of parameter types, across all trophic levels, and acting on different time and space scales. In that optimisation many of the surface parameters were difficult to constrain, because of a misfit function that consists mostly of observations in the deep ocean. Optimisation MOPS$^{\mathrm{oD}}$ presented here applies the same metric, but focuses on parameters in subsurface waters. The selection of parameters to be optimised is motivated by the large uncertainty regarding extent and expansion of oxygen minimum zones in models (Cocco et al., 2013; Cabre et al., 2015), and because little knowledge exists about their values, or even parameterisations.

Parameter $K_{\mathrm{O2}}$ determines the affinity of the aerobic remineralisation to oxygen, and the gradual transition from this process to denitrification (see Eqns. 15 and 20 of Kriest and Oschlies, 2015). $K_{\mathrm{DIN}}$ determines the affinity of denitrification to nitrate. Parameter $DIN_{\mathrm{min}}$ defines the lower threshold for the onset of denitrification. MOPS$^{\mathrm{oD}}$ also optimises the maximum rate of nitrogen fixation, $\mu_{NFix}^{*}$, which balances fixed nitrogen loss through denitrification. The fifth and sixth parameter to be estimated are the oxygen requirement per mole phosphorus remineralised, $R_{-O2:P}$, and the flux (or remineralisation) length scale, $b$. Upper and lower boundaries of parameters to be optimised have been set to a rather wide range (Table 1), to allow optimisation to explore a wide range of potential parameters. The optimal parameters of MOPS$^{\mathrm{oS}}$ for light and nutrient affinity

of phytoplankton, zooplankton grazing and its mortality are retained in MOPS$^{oD}$ (Table 1). Therefore optimisation MOPS$^{oD}$ builds upon a previous tuning of surface processes.

Most of the processes affected by the parameters to be optimised take place in suboxic waters, e.g. of the eastern equatorial Pacific (EEP). Given the coarse model geometry, it is possible that circulation dynamics are not represented well in the model. To investigate the influence of observations within this region on misfit function and parameter estimates, MOPS$^{oD}$ is repeated with a reduced data set, that excludes the EEP (here: east of $140°$W, between $10°$S and $10°$N) from the misfit function. This optimisation is named MOPS$_*^{oD}$.

## 2.7   Optimisation of RetroMOPS

In model RetroMOPS processes such as grazing of phytoplankton, and its subsequent release of organic or inorganic phosphorus are parameterised via a single component, DOP. Because DOP production and decay regulate the partitioning between sinking and dissolved organic matter, optimisation RetroMOPS$^o$ targets at these parameters, namely $\sigma$, $\lambda_s$ and $\lambda$. While $\sigma$, as parameter that regulates the export ratio, may be more or less well constrained, $\lambda_s$ and $\lambda$ both include a variety of processes, which may act on time scales of days to years. Hopkinson et al. (2002) applied a multi-G model to incubations of DOP sampled in surface waters of the middle Atlantic Bight, and measured decay constants for the very labile fraction (32% of total DOP) of $\approx 80$ y$^{-1}$, with a range of 3-254 y$^{-1}$. Half of total DOP was in the labile fraction and characterised by a decay constant of $\approx 7$ y$^{-1}$, ranging from 0.8-43 y$^{-1}$. However, these observations may not be directly transferable to globally simulated DOP, because most of the simulated ocean is far off the productive shelf areas; further, DOP in RetroMOPS is assumed to mimic a variety of biogeochemical components and processes. In a three-step optimisation study Letscher et al. (2015), who optimised a global model of semi-labile and refractory DOM against observations estimated rates of 0.016 y$^{-1}$ for semilabile DOP at the surface, and 0.22 y$^{-1}$ for semilabile DOP in the mesopelagial, i.e. much lower than suggested by Hopkinson et al. (2002). Summarising, the potential decay rate of the very labile to semi-labile fraction varies over several orders of magnitude, from $O(0.01) - O(100)$ y$^{-1}$.

Optimisation of RetroMOPS focuses on the dominant labile to semi-labile fraction, but allows for some potential fast turnover rates of DOP at the sea surface (towards the values observed by Hopkinson et al., 2002). To obtain a first impression on model sensitivity towards these parameters, a set of nine a priori experiments, that vary $\lambda$ between 0.18 y$^{-1}$ and 0.72 y$^{-1}$ and $\lambda_s$ between 0 y$^{-1}$ and 0.36 y$^{-1}$ has been carried out (Table 2), which provide a guidance for upper and lower boundaries for optimisation of RetroMOPS. To nevertheless explore the full range of potential decay rates, the maximum possible rate $(\lambda + \lambda_s)$ for optimisation is set to 7.2 y$^{-1}$, towards the average decay rate of the labile DOP observed by Hopkinson et al. (2002). Optimised RetroMOPS$^o$ will be compared to the sensitivity experiment with the lowest misfit ($\lambda_s = 0$, $\lambda = 0.36$), which is denoted as RetroMOPS$^r$.

The explicit representation of detritus in MOPS may result in considerable numerical diffusion (particularly on coarse vertical grids as used here; see also Kriest and Oschlies, 2011) and thus in a different estimate of optimal $b$ then when applying a direct flux curve, such as in RetroMOPS. Therefore, $b$ is included as fourth parameter to be optimised. The effect of explicit vs. implicit flux description on parameter estimate will be discussed in more detail below.

All other parameters (primary production, oxidant-dependent remineralisation, stoichiometry) have been fixed to those obtained in optimisations MOPS[oS] and MOPS[oD] (Table 1). By doing so, optimisation RetroMOPS[o] builds upon previous optimisations of the more complex MOPS, and overlooks the faint possibility that a parameter that is insensitive in one model, might not be so in another. While it might be desirable to optimise all parameters of RetroMOPS at once, this study rather aims at investigating to what extent a simpler model can serve as a shortcut to the more complex one, given the applied misfit function and observations.

## 3 Results and discussion

### 3.1 Optimal remineralisation parameters of MOPS

Both $R_{-O2:P}$ and $b$ are constrained very well by the observations, as indicated by a well-defined minimum of the misfit function (Fig. S3), and a narrow, almost gaussian distribution of the best 10% to 1% of parameters (Fig. 1). On the other hand, parameters related to the oxidant-affinity of remineralisation or nitrogen fixation are determined with lower accuracy. This is also reflected in the rather wide range of candidate solutions within 1‰ of the best misfit, which vary between 10% to 20% of their assigned a priori range (Table 3). Thus, in the presence of noise inherent in the observations, some parameters could only be estimated within a quite wide range of uncertainty, a feature that has already been addressed in a one-dimensional model by Löptien and Dietze (2015). So far, the potential consequences of this parametric uncertainty for other metrics (such as extent of oxygen minimum zones, OMZ) and possibly transient scenarios (e.g., their impact on simulated future evolution of OMZ volume) are not known.

The good determination of $b$ by dissolved inorganic tracers is in agreement with earlier studies that applied the same model (Kriest et al., 2017; Schartau et al., 2017). Its optimal value is very close to that obtained in MOPS[oS], i.e. higher than the value estimated by Kwon and Primeau (2006). Optimisation of maximum nitrogen fixation rate shows a slightly skewed distribution, but suggests an overall good estimate of this parameter. Optimal parameters for oxidant-dependent remineralisation also show wide, skewed distributions, with their mode near the lower ($K_{O2}$) or upper ($K_{DIN}$, $DIN_{min}$) boundary.

The high thresholds for the limitation of denitrification protect nitrate from becoming depleted in the upwelling regions, particularly the eastern equatorial Pacific, and resemble results obtained by Moore and Doney (2007): To prevent their model from reproducing unrealistically low nitrate values in this region, they had to impose a threshold of 32 mmol $NO_3$ m$^{-3}$ for the occurrence of denitrification. An explanation for this requirement of a high nitrate threshold might be found in the representation of the equatorial intermediate current system in coarse resolution models, which can result in spurious zonal oxygen gradients (Dietze and Loeptien, 2013; Getzlaff and Dietze, 2013). It is possible that the optimisation of biogeochemical parameters attempts to ameliorate these effects, which are in fact caused by the parameterisation of physics.

To further investigate the impact of this region on the parameter estimate, an additional optimisation was carried out, that targets at the same set of parameters, but omits the eastern equatorial Pacific from the calculation of the misfit function. This optimisation MOPS$_*^{oD}$ generates a lower threshold of nitrate for the onset of denitrification, and a higher maximum nitrogen fixation rate (Table 3), resulting in slightly enhanced fixed nitrogen turnover, particularly in the eastern equatorial Pacific

(Fig. 2). Compared to MOPS$^{\text{oD}}$ the estimates of $K_{\text{DIN}}$ and $DIN_{\text{min}}$ become more uncertain with respect to the best 10% to 1‰ individuals, and even show a bimodal distribution (Fig. S4, Table 3). The uncertainty in parameter estimates can be related to the missing data in regions of simulated denitrification. Because the misfit function excludes the EEP it is lower then when considering the entire ocean (Table 3). A posteriori evaluation of misfit to the entire data set results in a misfit of 0.439, the

same as for MOPS$^{\text{oD}}$. The only moderate effect of the eastern equatorial Pacific on optimization is likely related to the small volume occupied by this region, compared to total ocean volume.

Global fixed nitrogen turnover depends on parameters for oxidant-dependency of remineralisation: In MOPS$^{\text{oS}}$, both denitrification and nitrogen fixation are very high (Fig. 2), and outside the observed range (Table 4). Because of the reduced affinity to nitrate, in MOPS$^{\text{oD}}$ pelagic fixed nitrogen loss is almost halved, and now agrees with observed global estimates (Table 4).

Further, as a result of lower denitrification, the nitrate deficit in the eastern equatorial Pacific is smaller, but at the cost of a small underestimate of observed oxygen in this region (Fig. 3). The latter is a consequence of the now very low half-saturation constant for oxygen uptake (Table 3). In MOPS$_*^{\text{oD}}$ the constraint on nitrate affinity is again relaxed, resulting in an enhancement of fixed nitrogen turnover by about 20%, towards the upper limit of observed estimates (Table 4).

Overall, optimising parameters related to the oxidant affinity of oxic and suboxic remineralisation leads to a slightly im-

proved fit to tracer concentrations, to $J^* = 98\%$ of that of MOPS$^{\text{oS}}$ (Table 3), and to a better agreement with observed estimates of global biogeochemical fluxes (Table 4). Although the eastern equatorial Pacific, and potential unresolved processes in simulated circulation, has no effect on global misfit, its effect on some parameter estimates, however, results in an increase in global fixed nitrogen loss of about 20%.

## 3.2    A shortcut for surface biology: RetroMOPS

Given that parameters related to surface biology were difficult to constrain in MOPS$^{\text{oS}}$, and, within a certain range, exert only a small influence on the fit to global tracer distributions (Kriest et al., 2017), this section examines if RetroMOPS, as a model that parameterises surface biology in a much simpler way, suffices to represent biogeochemical tracer fields. Starting from growth and decay parameters optimised in MOPS, sensitivity experiments and optimisation search for optimal parameters for DOP production and decay, that mimic the surface nutrient turnover of MOPS.

### 3.2.1    Sensitivity to DOP production and decay

In RetroMOPS fast DOP recycling results in higher primary production, export production, and deep organic particle flux, especially in the equatorial upwelling regions (Fig. 4). While this has only a small effect on vertically or globally averaged phosphate concentrations (Figures 5 and 6), it causes a large underestimate of nitrate in the ocean (Figures S6 and 6). The underestimate can be explained by the tight coupling between production, export and denitrification, which leads to higher

denitrification and global fixed N-loss (Fig. 4), and thus a larger nitrate deficit (Fig S6) in the eastern equatorial Pacific, in agreement with effects hypothesised and investigated by Landolfi et al. (2013).

In contrast, nitrogen fixation is not much affected by DOP turnover rates. The imbalance between nitrogen losses and gains suggests that the models even after 3000 years of simulation are not yet in equilibrium, which might be explained by the large

spatial scales between regions of fixed nitrogen loss and gain. The divergence increases with higher DOP recycling rates (and thus larger denitrification), indicating that there is no unique equilibration time scale for one and the same model, but that it depends on biogeochemical parameters associated with sinking and remineralisation of organic matter, as observed earlier (Kriest and Oschlies, 2015). The resulting requirement for long spin-up times for a complete model adjustment, their dependence on biogeochemical parameters, and the model's nonlinearity during spin-up (Kriest and Oschlies, 2015), complicate model calibration and assessment, in addition to those factors already investigated by Seferian et al. (2016). It emphasises the need for a thorough assessment of trade-offs between model complexity and computational demand, and the possibility to examine the parameter space in sufficient detail.

The effect of DOP recycling on oxygen concentrations differs from its effect on nitrate. With fast recycling DOP is remineralised mostly at its place of production, and does not contribute much to oxygen consumption in deep waters (see also Fig S5). As a consequence, deep oxygen concentrations are high, particularly in the northern North Pacific (Fig. 5), and global average oxygen is overestimated by more than 10% (Fig. 6). Slow DOP recycling, in contrast, leads to less organic matter remineralisation in well-ventilated waters, but more remineralisation in deep waters. This in turn results in an underestimate of global mean oxygen of almost 10% (for $\lambda = 0.18$ y$^{-1}$ and $\lambda_\mathrm{s} = 0$ y$^{-1}$), which is somewhat surprising, given that production and export in this scenario are the lowest of all simulations (Fig. 4). Overall, the best fit to observed inorganic tracer concentrations is achieved with moderate DOP recycling (Table 2, Fig. 5).

Most likely because of its fixed inventory, phosphate contributes to less than 1/3 of the misfit function, and is quite insensitive to changes in DOP recycling rate (Fig. 6). Nitrate and oxygen play a larger role for model fit, because their inventory can adapt to changing biogeochemistry. The misfit to nitrate and oxygen increases more or less in concert with their bias (Fig. 6). Therefore, these tracers with their flexible inventory provide some very useful constraints on DOP recycling rates.

Slow DOP recycling increases DOP concentrations at the surface, particularly in the ACC and in the northern North Atlantic (Fig. 5) towards concentrations that exceed the observations (Yoshimura et al., 2007; Raimbault et al., 2008; Torres-Valdes et al., 2009; Letscher and Moore, 2015). Only the simulation with quite fast DOP recycling of $\lambda = 0.72$ y$^{-1}$ and $\lambda_\mathrm{s} = 0.36$ y$^{-1}$ results in reasonable concentrations of DOP - but at the cost of too high phosphate concentrations along these sections, and a too high global misfit (Table 2), a too low nitrate and too high oxygen inventory (Figures 5 and 6). Therefore, it should be noted that despite the relatively good fit of RetroMOPS$^\mathrm{r}$, it nevertheless suffers from a potential mismatch to DOP, which so far is not included in misfit evaluation.

### 3.2.2 Optimal parameters for DOP cycling in RetroMOPS

All four parameters of RetroMOPS$^\mathrm{o}$ are well constrained by the observations, as indicated by the narrow, almost gaussian distribution around the optimal parameter (Figures 7, S7, and Table 3). Optimisation reduces the decay rate for surface DOP, $\lambda_\mathrm{s}$, to almost zero, i.e., in RetroMOPS there seems to be no requirement for fast DOP turnover at the surface, similar to the results obtained by Letscher et al. (2015). The optimal total remineralisation rate of DOP $(\lambda + \lambda_\mathrm{s})$ is about 0.5 y$^{-1}$, more than twice as high as the recycling rate estimated by Letscher et al. (2015), but lower than the rates observed by Hopkinson et al.

(2002). The optimal fraction of primary production released as DOP , $\sigma$, is 73% and agrees very well with $\sigma = 0.74$ obtained by Kwon and Primeau (2006); however, their optimal DOP decay rate was twice as high (1 y$^{-1}$).

When optimising a simple biogeochemical model similar to RetroMOPS against observed phosphate, Kwon and Primeau (2006) noted a correlation between DOP production fraction and decay rate, impeding the simultaneous estimation of these parameters. On the contrary, in optimisation RetroMOPS$^{\text{o}}$ both $\sigma$ and the DOP decay rates seem to be rather well constrained. An analysis of the different components of the misfit function, similar to Fig. 4 of Kwon and Primeau (2006), helps to resolve this apparent contradiction. For this, in Fig. 8 the total misfit $J$ and its components $J(j)$ of Eqn. 11, as well as the bias of the best 5% of all individuals are mapped against $\sigma$ and DOP decay timescale $\tau = 1/(\lambda + \lambda_{\text{s}})$.

Note that the analysis depicted in Fig. 8 differs from that of Kwon and Primeau (2006) in several aspects: Firstly, their global biogeochemical model was fully equilibrated (due to their direct evaluation of steady state via Newton's method), whereas simulations of RetroMOPS may still exhibit some drift in nitrogen inventory (see subsection 3.2.1 and supplement). Second, Kwon and Primeau (2006) evaluated model sensitivity at $b = 1$, while Fig. 8 displays a region $\pm 5\%$ around optimal $b = 0.98$. Thirdly, Fig. 8 maps only the misfit of solutions realised by the optimisation routine, while Kwon and Primeau (2006) analysed the entire parameter space at $b = 1$. Most important, the misfit function applied here is based on three components, with very different properties and associated time scales (see above), which can be of advantage for parameter estimation.

The misfit to phosphate (Fig. 8, lower left panel) indicates an elongated valley in the two-dimensional projection on DOP decay timescale $\tau$ (years) and DOP production fraction $\sigma$, and resembles Fig. 4 of Kwon and Primeau (2006). Indeed, one of the lowest misfits to phosphate is achieved with about the same set of parameters as in Kwon and Primeau (2006), namely $\tau \approx 1$, $\sigma \approx 0.73$. However, nitrate and oxygen show a different, and, partly, antagonistic, pattern: the best fit to observed nitrate is achieved with rather high values of $\sigma \approx 0.8$ and $\tau$ between about 1-2 years, while the best fit to oxygen is obtained with $\sigma \approx 0.7$ and $\tau \approx 1.5$ years. The superposition of the different components of the misfit function leads to a unique optimum at $\tau = 2$ ($\lambda = 0.47$ and $\lambda_{\text{s}} = 0.02$) and $\sigma = 0.73$ (Table 3). Thus, oxygen and nitrate can provide some useful, independent information on these parameters.

This can partly be explained by their non-conservative nature. As noted in section 3.2.1 the inventory of these tracers may change freely according to model parameterisation. The resulting bias to observations thus adds two important components to the misfit function, both of which are independent: while high DOP turnover (as simulated by low $\tau$) biases nitrate low (Fig. 8, upper mid panel), the same value leads to an overestimate of oxygen (Fig. 8, upper right panel; see also Fig. 6). This behaviour can be explained with the different processes and boundary conditions for the two tracers already noted in section 3.2.1: a high DOP turnover leads to higher fluxes and a tighter coupling of production and denitrification in upwelling waters, causing a nitrate deficit in the model (see above, and Fig S6). On the other hand, it reduces preformed DOP in subducted waters e.g., the Southern Ocean, thereby decreasing aerobic remineralisation and oxygen consumption in these waters on their passage towards, e.g., the northern North Pacific. The latter process increases oxygen particularly in deep waters (Fig S5).

To summarise, including nitrate and oxygen as non-conservative tracers in the misfit function helps to resolve parameters related to DOP production and decay on long time scales. This can be explained by the different pathways of DOP originating from upwelling regions or subducted water masses in the high latitudes, and is confirmed by the analysis of sensitivity exper-

iments presented in section 3.2.1. However, a better fit to observed phosphate seems to come at the expense of a mismatch to observed DOP concentration. It remains to be investigated, if a simultaneous fit to observed inorganic and organic phosphorus is possible.

### 3.2.3 Comparison of MOPS and RetroMOPS

The optimal $b = 0.98$ of RetroMOPS$^{\text{o}}$ is lower than that of MOPS$^{\text{oS}}$ and MOPS$^{\text{oD}}$. This may be partially explained with the absence of numerical diffusion of detritus in RetroMOPS. As shown by Kriest and Oschlies (2011), in models that explicitly simulate detritus sinking with an upstream scheme the assumption of homogenous detritus distribution in each vertical grid box causes an additional, usually downward transport of detritus. This results in an effective $b$ which is about 10-20% smaller (corresponding to faster sinking) than the nominally prescribed $b$. Optimisation of MOPS accounts for this additional numerical

transport by increasing $b$ (= reducing sinking velocity) by some amount. Therefore, optimal $b$ of MOPS without any influence numerical of diffusion would likely be around 1.1-1.2, i.e. closer to $b = 0.98$ of RetroMOPS$^{\text{o}}$. Considering this effect, optimal $b$ of MOPS$^{\text{oD}}$ and, in particular, RetroMOPS$^{\text{o}}$ agree with the optimal value of $b = 1$ found by Kwon and Primeau (2006).

   Despite its generally lower fluxes, fixed nitrogen loss in the eastern equatorial Pacific is higher in RetroMOPS$^{\text{o}}$ than in MOPS$^{\text{oD}}$ (Fig. 2), resulting in a nitrate deficit in this region. Likely, the instantaneous remineralisation of sinking material

inherent in the direct flux parameterisation of RetroMOPS causes a tighter spatial coupling between production, sinking, remineralisation and upwelling (see also section 3.2.1). It has been suggested earlier that the production of slowly degradable organic matter above upwelling regions and/or oxygen minimum zones may help to decouple these processes, and avoid a runaway effect of nitrate loss (Landolfi et al., 2013; Dietze and Loeptien, 2013). The very low optimal value for surface DOP turnover $\lambda_{\text{s}}$ found in this study, and also in the study by Letscher et al. (2015) supports this finding.

Simulated biogeochemical fluxes of RetroMOPS$^{\text{o}}$ are generally lower than those of MOPS$^{\text{oD}}$, and their horizontal pattern is less pronounced (Fig. 2). This likely arises from the prescribed, constant phytoplankton concentration of RetroMOPS$^{\text{o}}$, which mutes biogeochemical dynamics in productive regions of the high latitudes and upwelling areas. Because RetroMOPS$^{\text{o}}$ applies the same parameters as MOPS$^{oD}$ for oxidant-dependent processes, its global fixed nitrogen loss and gain is comparable to that of the more complex model.

The total misfit to observed dissolved tracer concentrations of RetroMOPS$^{\text{o}}$ is only about 4% higher than that of MOPS$^{\text{oD}}$, suggesting that even the simple RetroMOPS can perform almost as well as MOPS with respect to annual mean phosphate, nitrate, and oxygen. As for MOPS, optimisation of RetroMOPS against dissolved tracer concentrations results in a good fit to global estimates of biogeochemical fluxes (Table 4), and indicates that these tracers can provide means to calibrate biogeochemical model fluxes on a global scale, even - or especially - for a model as simple as RetroMOPS.

### 3.3 How much complexity is needed?

Current, state-of-the-art biogeochemical models address questions such as the future evolution of oxygen minimum zones, or uptake of anthropogenic carbon by the ocean (e.g. Cocco et al., 2013; Cabre et al., 2015; Kwiatkowski et al., 2014). Compared to these models MOPS and RetroMOPS presented here are of a rather low structural complexity. RetroMOPS is quite similar

to early models addressing these tasks, among them the pioneering work of Ernst-Maier Reimer (e.g. Bacastow and Maier-Reimer, 1990, 1991; Maier-Reimer, 1993), while MOPS resembles models of intermediate complexity such as HAMOCC (e.g. Six and Maier-Reimer, 1996; Maier-Reimer et al., 2005) or HadOCC (Palmer and Totterdell, 2001). However, very simple models such as RetroMOPS are still being used, e.g., for inverse methods (e.g. Kwon and Primeau, 2006, 2008) or to investigate specific processes, where their computational efficiency and structural simplicity facilitates model analysis (e.g. Parekh et al., 2005; Kwon et al., 2009; Primeau et al., 2013). In contrast to these very simple model are models that simulate different plankton groups and size classes of detritus, e.g. PISCES (Aumont et al., 2015), MEDUSA (Yool et al., 2013), or PlankTOM (Le Quere et al., 2005).

Despite this large range of structural complexity, there have been only few studies which evaluate these models against a common data set, and with a common circulation. One example is the study by Kwiatkowski et al. (2014), who compared the output of six different global biogeochemical models, coupled to a common circulation model, and simulated over 118 years, against data sets of surface $pCO_2$, DIC, alkalinity, DIN, Chl $a$ and primary production. The models varied in complexity from seven to 57 compartments, and thus also in their computational demand by almost a factor of five. To assess model skill Kwiatkowski et al. (2014) ranked the models with respect to spatial correlation between, and variance of, model and observations. In general, the more complex models performed better with respect to simulated variance, but the simpler models better with respect to spatial correlation. Although no model was superior across all metrics, they concluded that 'Results suggest little evidence that higher biological complexity implies better model performance in reproducing observed global-scale bulk properties of ocean biogeochemistry." (Kwiatkowski et al., 2014).

The lack of distinction between models and their ability to represent biogeochemical tracers is corroborated by the study by Galbraith et al. (2015), who evaluated three different biogeochemical ocean models within a common framework for the earth system. The models varied in complexity between one to 30 components. Following a spin-up over 100 years, Galbraith et al. (2015) analysed both a transient and preindustrial scenario with respect to the model's representation of macronutrients, oxygen, DIC, and export. All three models performed quite similar with respect to the observed tracer fields, as well as with the transient evolution of carbon uptake and oxygen concentrations. Therefore, in the presence of noise inherent in observations, and given the sparsity of biological data sets, so far it seems unresolved if more complexity is indeed beneficial - at least if the model is supposed to represent mostly biogeochemical processes, instead of biological interactions, and is compared against bulk biogeochemical properties.

## 4 Conclusions

Based on a global metric for biogeochemical tracers this study assessed the skill of two optimised global biogeochemical ocean models, as well as the metric's capability to constrain the often uncertain model parameters.

Similar to an earlier study (Kriest et al., 2017) that targeted as parameters relevant for biogeochemical processes at the sea surface, parameters for oxidant-dependent processes in the mesopelagic could only be determined with a wide range of uncertainty. The reason for this lack of resolution can be found in the small volume occupied by either surface, or oxygen

minimum zones (where oxidant-dependency is of relevance). Omission of the eastern equatorial Pacific from the misfit function increases uncertainty in parameter estimates, but does not fundamentally alter the outcome of optimisation, likely because of the small volume of this region.

In contrast, parameters relevant for large-scale, global distributions of oxygen, such as remineralisation length scale or stoichiometry could be determined well; these parameters were very similar in all experiments, and point towards a shorter remineralisation length scale of $b = 1.3$ to $b = 1.4$, as compared to the canonical $b = 0.858$ suggested by Martin et al. (1987).

Despite the uncertainty in estimates of some parameters, and very small differences between models in the residual misfit, optimisation of parameters for oxidant-dependent processes results in a much better fit to observed estimates of global fixed nitrogen turnover. The remaining mismatch to observations can partly be attributed to circulation. Model optimisations with different parameterisations of circulation and the equatorial intermediate current system (e.g., using TMs extracted from the UVic model; Kvale et al., 2017) will help to examine, if a different parameterisation alters the current requirement for very high nitrate threshold of denitrification, that currently helps to prevent nitrate from depletion.

Oxygen and nitrate add important additional constraints on the estimation of biogeochemical parameters. Of particular importance is that, in addition to the spatial information they provide, their flexible inventory introduces the bias as additional information for model calibration. The different time and space scales of processes relevant for their inventory may help to constrain parameters that govern dissolved organic matter production and decay. The effect of these tracers on parameter estimates is of particular importance for models such as RetroMOPS and MOPS, that aim at conserving all oxidants. It may be weaker for models that continue remineralisation even under suboxic and/or low nitrate conditions, thereby implicitly assuming some "hidden" oxidants. In these models it could be useful to track and examine potential oxidant deficits for model evaluation.

DOP recycling rate affects surface DOP and phosphate concentrations conversely: either the model performs relatively well with respect to DOP. In this case phosphate concentrations are overestimated by the model. If the model performs well with respect to phosphate, it overestimates surface DOP. Observations of DOP as additional constraint on model parameters will help to find out if there is a model solution that fits all tracers equally well.

With respect to annual mean tracer concentrations the simple model RetroMOPS can perform almost as well as the more complex model MOPS, the residual misfit being only 5% larger. Spatial patterns of fluxes in RetroMOPS are less pronounced, but global tracer concentrations, inventories and fluxes are comparable to that of MOPS, and in agreement with observed estimates.

Although it is obvious that low to intermediate complexity models such as the models presented here cannot represent the level of detail embedded in models with e.g., several plankton size classes, so far evaluation with respect to the bulk, biogeochemical observations does not seem to indicate any superiority of more complex models on a global scale. This of course may change if our scientific interest and model purpose is directed towards shorter time scales, or surface patterns, for which the misfit function applied provides little information. In this case more complex data sets, such as different plankton groups, or particle size distribution, may provide further insight about the level of model complexity required. If focusing on large scales, however, a simple model such as RetroMOPS or similarly simple models may suffice to represent and analyse much of the biogeochemical dynamics in the ocean.

*Acknowledgements.* I am very thankful for having met Ernst Maier-Reimer, who pioneered global biogeochemical modelling. In his thoughtful and kind way he taught me to view global ocean biogeochemistry before the background of long time and large space scales.

This work is a contribution to the DFG-supported project SFB754 and to BMBF joint project PalMod (FKZ 01LP1512A). Parallel supercomputing resources have been provided by the North-German Supercomputing Alliance (HLRN). The author wishes to acknowledge use of the Ferret program of NOAA's Pacific Marine Environmental Laboratory for analysis and graphics in this paper. I thank three anonymous reviewers and Friederike Hoffmann for their constructive and helpful comments.

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

**Table 1.** Experimental setup of optimisation. Parameters that stay fixed are highlighted. For parameters subject to optimisation we indicate the assigned, a priori lower and upper parameter boundary (parameter range, $R_\ominus^A$) for optimisation in square brackets. "-": not applicable for this model.

| Experiment | MOPS$^r$ | MOPS$^{oS}$ | MOPS$^{oD}$ | RetroMOPS$^r$ | RetroMOPS$^o$ | unit |
|---|---|---|---|---|---|---|
| $\sigma$ | - | - | - | **0.67** | [0.4 - 0.8] | |
| $\lambda_s$ | - | - | - | **0** | [0.0 - 3.6] | y$^{-1}$ |
| $\lambda$ | **0.17** | **0.17** | **0.17** | **0.36** | [0.036 - 3.6] | y$^{-1}$ |
| $I_c$ | **24** | [4 - 48] | **9.65** | **9.65** | **9.65** | W m$^{-2}$ |
| $K_{PHY}$ | **0.03125** | [0.001 - 0.5] | **0.5** | **0.5** | **0.5** | mmol P m$^{-3}$ |
| $\mu_{ZOO}$ | **2** | [1 - 3] | **1.89** | - | - | d$^{-1}$ |
| $\kappa_{ZOO}$ | **3.2** | [1.6 - 4.8] | **4.55** | - | - | (d mmol P m$^{-3}$)$^{-1}$ |
| $b^*$ | **0.858** | [0.4 - 1.8] | [0.4 - 1.8] | **1.0725** | [0.4 - 1.8] | |
| $R_{-O2:P}$ | **170** | [150 - 200] | [150 - 200] | **171.7** | **171.7** | mmol O$_2$:mmol P |
| $\mu_{NFix}$ | **2** | **2** | [1 - 3] | **1.19** | **1.19** | nmol N d$^{-1}$ |
| $DIN_{min}$ | **4** | **4** | [1 - 16] | **15.80** | **15.80** | mmol N m$^{-3}$ |
| $K_{O2}$ | **2** | **2** | [1 - 16] | **1.0** | **1.0** | mmol O$_2$ m$^{-3}$ |
| $K_{DIN}$ | **8** | **8** | [2 - 32] | **31.97** | **31.97** | mmol N m$^{-3}$ |

$^*$ Note that from $b$ (the optimised parameter) in MOPS we calculate the rate of vertical increase in sinking speed $a$ of $w = a\,z$, via $a = r/b$. For $r$ we assume nominal detrital remineralisation of $r = 0.05$ d$^{-1}$. The resulting values for $a$ are: 0.058275 ($b = 0.858$), 0.0278 (lower boundary) and 0.125 (upper boundary).

**Table 2.** Results (misfit $J$) of sensitivity experiments with model RetroMOPS, regarding parameters $\lambda_s$ and $\lambda$ for DOP decay rate. The misfit of the reference scenario RetroMOPS$^r$ is indicated in bold.

| | $\lambda_s = 0$ | $\lambda_s = 0.18$ | $\lambda_s = 0.36$ |
|---|---|---|---|
| $\lambda = 0.18$ | 0.502 | 0.480 | 0.480 |
| $\lambda = 0.36$ | **0.466** | 0.476 | 0.493 |
| $\lambda = 0.72$ | 0.503 | 0.522 | 0.539 |

**Table 3.** Optimisation results: minimum misfit $J^*$, optimum parameters and their uncertainties. To determine parameter uncertainty, we selected a group $\Omega$ of the 1‰ best individuals, i.e. individuals defined by a misfit $J_i : J_i/J^* - 1 \leq \Delta_J$, with $\Delta_J = 0.001$. The number of these individuals $N(\Omega)$ is also denoted as fraction $n(\Omega)$ of all individuals of the optimisation $\lambda \times N$, where $N$ is the number of generations, and $\lambda = 10$ the population size. For each parameter $\Theta$ the first column gives the optimal parameter $\Theta^*$ (i.e., the average parameter of the last generation). The second and third column present the parameter range of all individuals of $\Omega$, expressed as absolute value ($R_\Theta(\Omega)$), and normalised by the a priori range of parameters ($R_\Theta^A$; see Table 1): $r_\Theta(\Omega) = R_\Theta(\Omega)/R_\Theta^A$ value.

| Experiment: | MOPS$^{\mathrm{oS}}$ | | | MOPS$^{\mathrm{oD}}$ | | | MOPS$^{\mathrm{oD}}_*$ | | | RetroMOPS$^{\mathrm{o}}$ | | |
|---|---|---|---|---|---|---|---|---|---|---|---|---|
| Parameter | $\Theta^*$ | $R_\Theta(\Omega)$ | $r_\Theta(\Omega)$ | $\Theta^*$ | $R_\Theta(\Omega)$ | $r_\Theta(\Omega)$ | $\Theta^*$ | $R_\Theta(\Omega)$ | $r_\Theta(\Omega)$ | $\Theta^*$ | $R_\Theta(\Omega)$ | $r_\Theta(\Omega)$ |
| $\sigma$ | - | - | - | - | - | - | - | - | - | 0.73 | [0.7-0.7] | 6 |
| $\lambda_{\mathrm{s}}$ | - | - | - | - | - | - | - | - | - | 0.02 | [-0.1-0.2] | 8 |
| $\lambda$ | - | - | - | - | - | - | - | - | - | 0.47 | [0.4-0.5] | 4 |
| $I_{\mathrm{c}}$ | 9.66 | [8.9-10.3] | 3 | | | | | | | | | |
| $K_{\mathrm{PHY}}$ | 0.50 | [0.4-0.5] | 28 | | | | | | | | | |
| $\mu_{\mathrm{ZOO}}$ | 1.89 | [1.6-2.0] | 22 | | | | | | | - | - | - |
| $\kappa_{\mathrm{ZOO}}$ | 4.57 | [3.0-4.7] | 53 | | | | | | | - | - | - |
| $b^{\S}$ | 1.34 | [1.3-1.4] | 4 | 1.39 | [1.4-1.4] | 3 | 1.41 | [1.4-1.4] | 2 | 0.98 | [1.0-1.0] | 2 |
| $R_{-\mathrm{O2:P}}$ | 167.0 | [165-170] | 9 | 171.7 | [170-173] | 6 | 174.9 | [174-176] | 5 | | | |
| $\mu_{\mathrm{NFix}}$ | | | | 1.19 | [1.1-1.4] | 13 | 1.47 | [1.4-1.6] | 10 | | | |
| $DIN_{\mathrm{min}}$ | | | | 15.80 | [13-16] | 20 | 12.96 | [12-16] | 25 | | | |
| $K_{\mathrm{O2}}$ | | | | 1.00 | [0.3-1.8] | 10 | 1.00 | [0.5-1.4] | 6 | | | |
| $K_{\mathrm{DIN}}$ | | | | 31.97 | [30-34] | 12 | 31.97 | [22-33] | 35 | | | |
| $J^*$ | | 0.450 | | | 0.439 | | | 0.427 | | | 0.458 | |
| $\lambda \times N$ | | 1820 | | | 1190 | | | 2000 | | | 660 | |
| $N(\Omega)$ | | 718 | | | 514 | | | 1285 | | | 262 | |
| $n(\Omega)$ | | 39 | | | 43 | | | 64 | | | 40 | |

**Table 4.** Global annual fluxes of primary production (P), grazing (GRAZ), fixed nitrogen loss through pelagic denitrification (NLOSS), export production (F120, flux through 120 m), flux through 2250 m (F2250), and benthic burial (BUR), in Pg N y$^{-1}$, for the reference experiment of MOPS$^r$, MOPS$^{oS}$, MOPS$^{oD}$, MOPS$_*^{oD}$ and RetroMOPS, for which we show the fluxes of the (best) reference experiment, RetroMOPS$^r$, the range of all sensitivity experiments, and the optimised run, RetroMOPS$^o$. Also shown are some globally derived, observed estimates. Conversion between different elements was carried out via N:P=16, and C:P=122.

| Experiment | P | GRAZ | NLOSS | F120 | F2250 | BUR |
|---|---|---|---|---|---|---|
| MOPS$^r$ | 5.44 | 3.52 | 0.098 | 0.918 | 0.107 | 0.051 |
| MOPS$^{oS}$ | 7.52 | 4.74 | 0.117 | 1.102 | 0.056 | 0.018 |
| MOPS$^{oD}$ | 7.70 | 4.97 | 0.068 | 1.080 | 0.055 | 0.022 |
| MOPS$_*^{oD}$ | 7.80 | 5.06 | 0.083 | 1.081 | 0.053 | 0.021 |
| RetroMOPS$^r$ | 5.56 | - | 0.078 | 1.194 | 0.043 | 0.010 |
| RetroMOPS (range) | 4.88-6.21 | - | 0.076-0.084 | 1.076-1.286 | 0.039-0.047 | 0.008-0.014 |
| RetroMOPS$^o$ | 6.31 | - | 0.071 | 1.12 | 0.052 | 0.009 |
| Observed[§] | 7.68-8.09 | 4.79-5.71 | 0.05-0.08 | 0.29-1.53 | 0.03-0.07 | 0.02 |

[§] Observed fluxes are from Carr et al. (2006, primary production), Honjo et al. (2008, particle flux), Lutz et al. (2007, particle flux), Dunne et al. (2007, particle flux), Schmoker et al. (2013, primary production, zooplankton grazing excluding/including mesozooplankton grazing), Wallmann (2010, burial; without shelf and slope region), and Kriest and Oschlies (2015, fixed nitrogen loss).

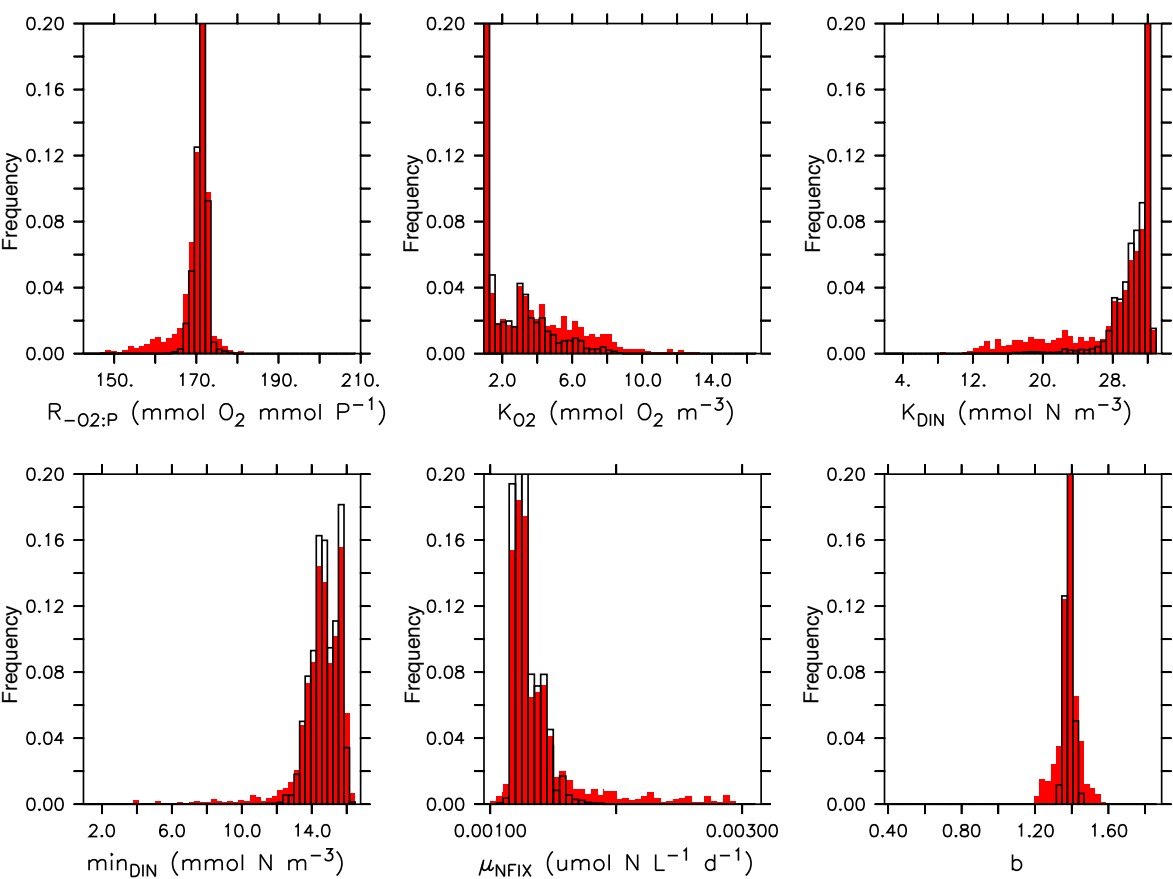

**Figure 1.** Parameter distribution of model simulations obtained during the optimisation of MOPS[oD], whose misfit do not exceed a threshold limit of $\Delta J = 1.1\,J^*$ (10%, red bars) or $\Delta J = 1.01\,J^*$ (1%, open bars) of the minimum misfit $J^*$. For the projection parameters of all model simulations in the optimisation trajectory were grouped into 50 classes.

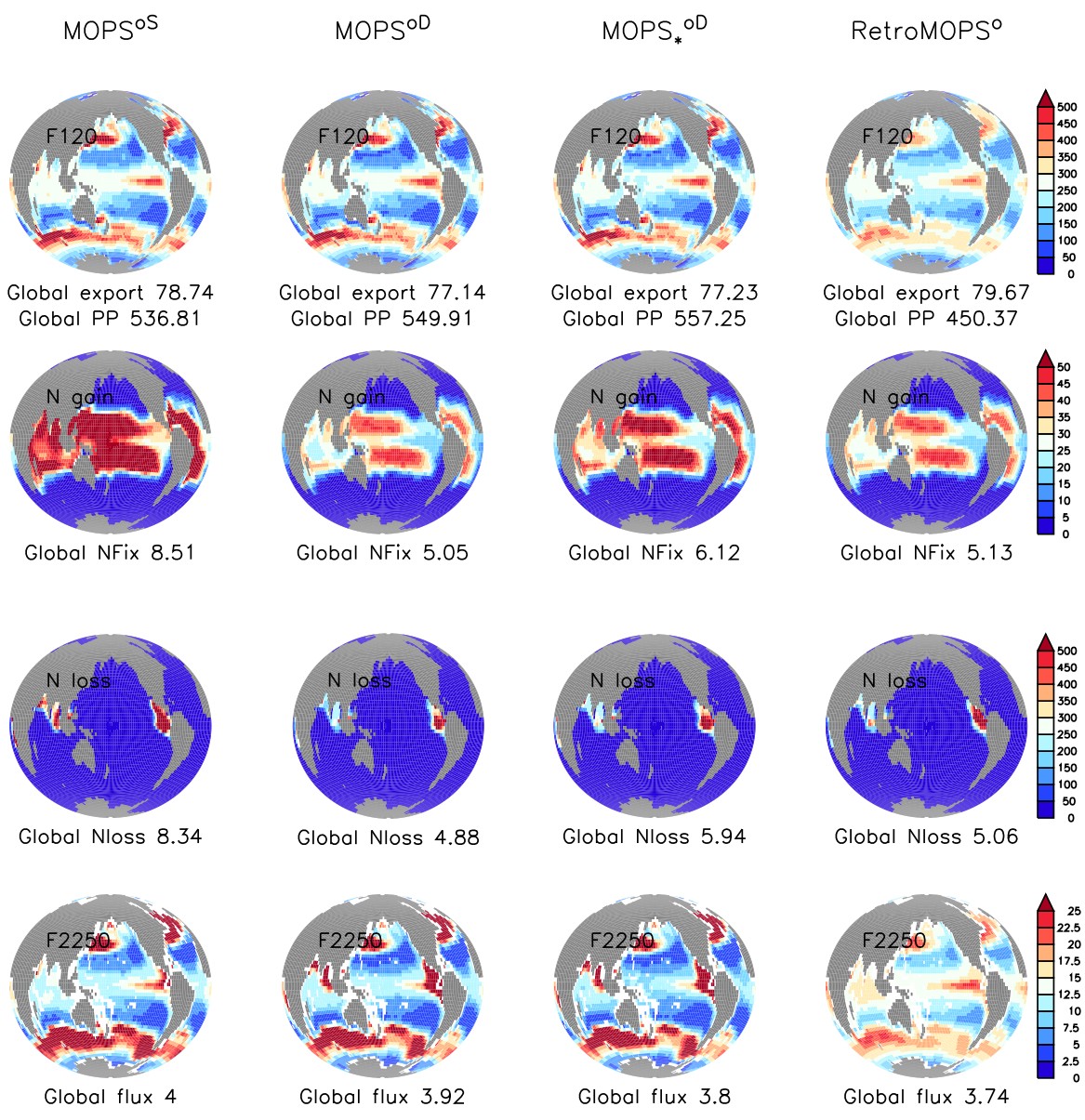

**Figure 2.** Biogeochemical fluxes of MOPS[oS], MOPS[oD], MOPS[oD]∗ and RetroMOPS[o]. Top: Export production (here: sedimentation at 120 m). Second row from top: nitrogen fixation. Third row from top: fixed nitrogen loss through pelagic denitrification. Bottom: sedimentation at 2250 m. All fluxes in mmol N m$^{-2}$ y$^{-1}$. Each subpanel also gives the global flux in Tmol N y$^{-1}$.

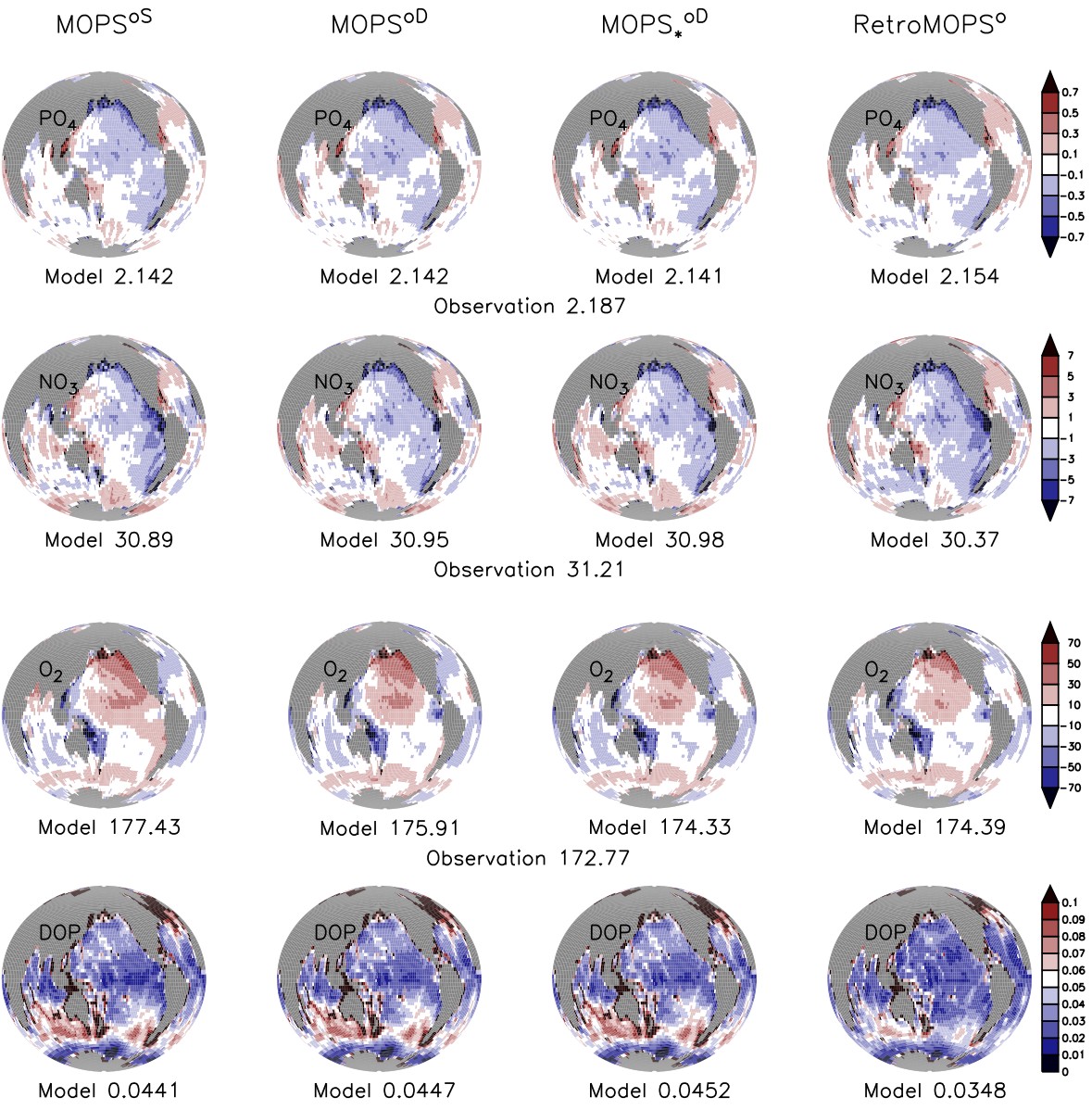

**Figure 3.** Vertically averaged tracers of MOPS[oS], MOPS[oD], MOPS[oD]_* and RetroMOPS[o]. Top: phosphate. Second row from top: nitrate. Third row from top: oxygen. Bottom: DOP. Phosphate (mmol P m$^{-3}$), nitrate (mmol N m$^{-3}$) and oxygen (mmol O$_2$ m$^{-3}$) are expressed as deviation from observations (Garcia et al., 2006a, b), DOP is given in absolute concentrations (mmol P m$^{-3}$). Each subpanel also gives the global average tracer concentration in mmol m$^{-3}$.

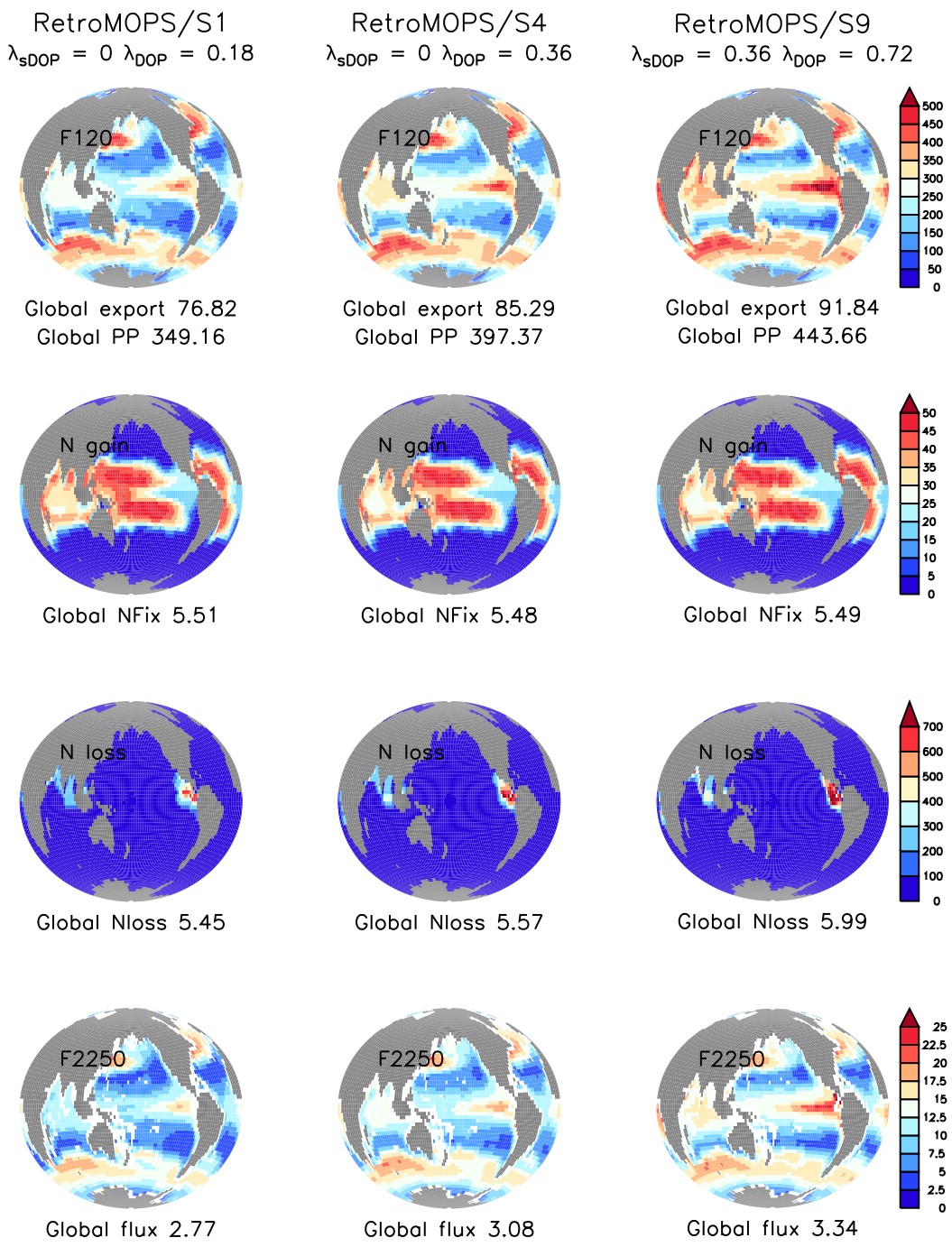

**Figure 4.** As Fig. 2, but for three sensitivity experiments with model RetroMOPS.

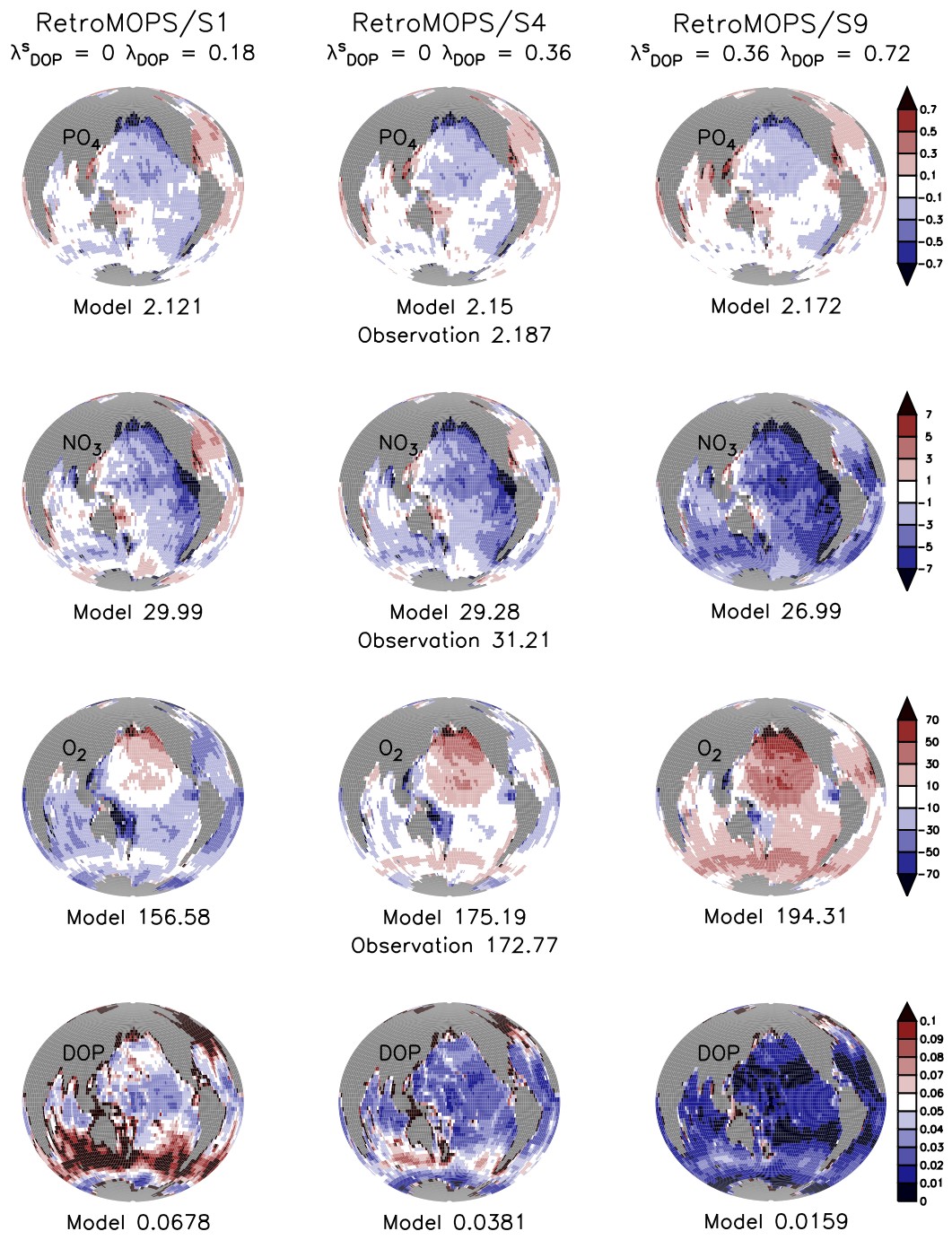

**Figure 5.** As Fig. 3, but for three sensitivity experiments with model RetroMOPS.

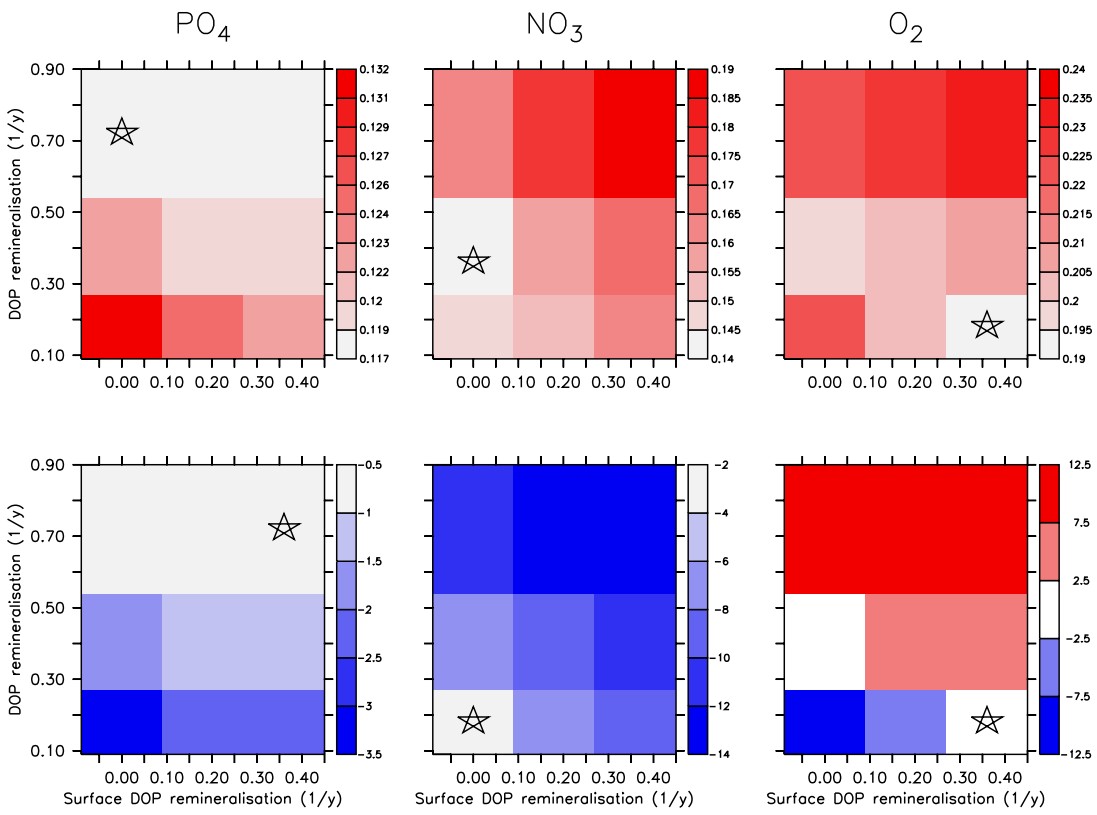

**Figure 6.** Components of the misfit function ($J(j)$ of Eqn. 11; upper panels) and model bias (lower panels), projected onto $\lambda_s$ and $\lambda$. Bias is expressed as $(\overline{m_j}/\overline{o_j} - 1) \times 100$, where $\overline{m_j}$ is the global average model tracer, and $\overline{o_j}$ the average observed tracer, for the three tracers phosphate ($j = 1$; left panels), nitrate ($j = 2$; mid panels) and oxygen ($j = 3$; right panels). An open star indicates the respective lowest misfit or bias.

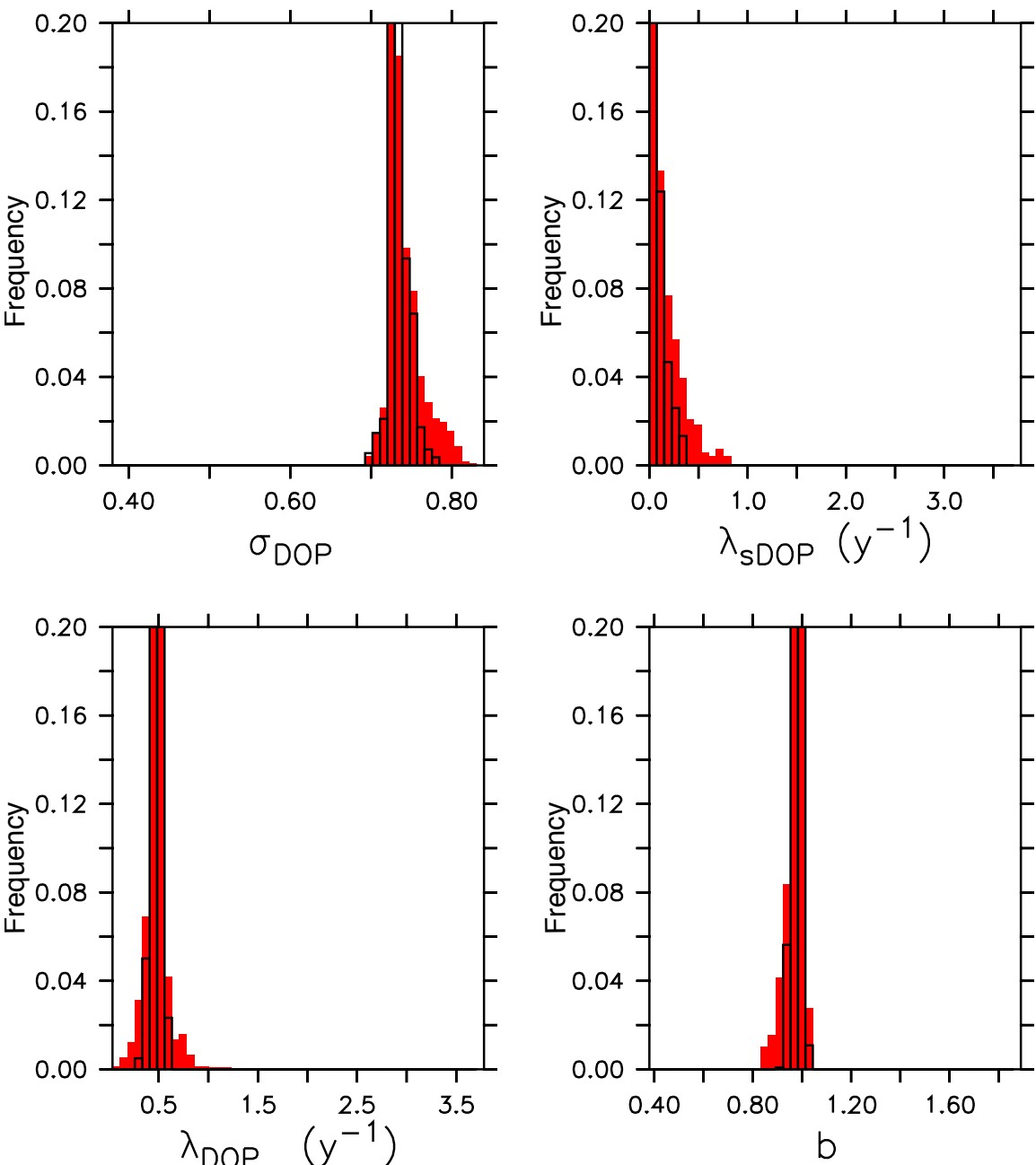

**Figure 7.** As Fig. 1, but for optimisation RetroMOPS°.

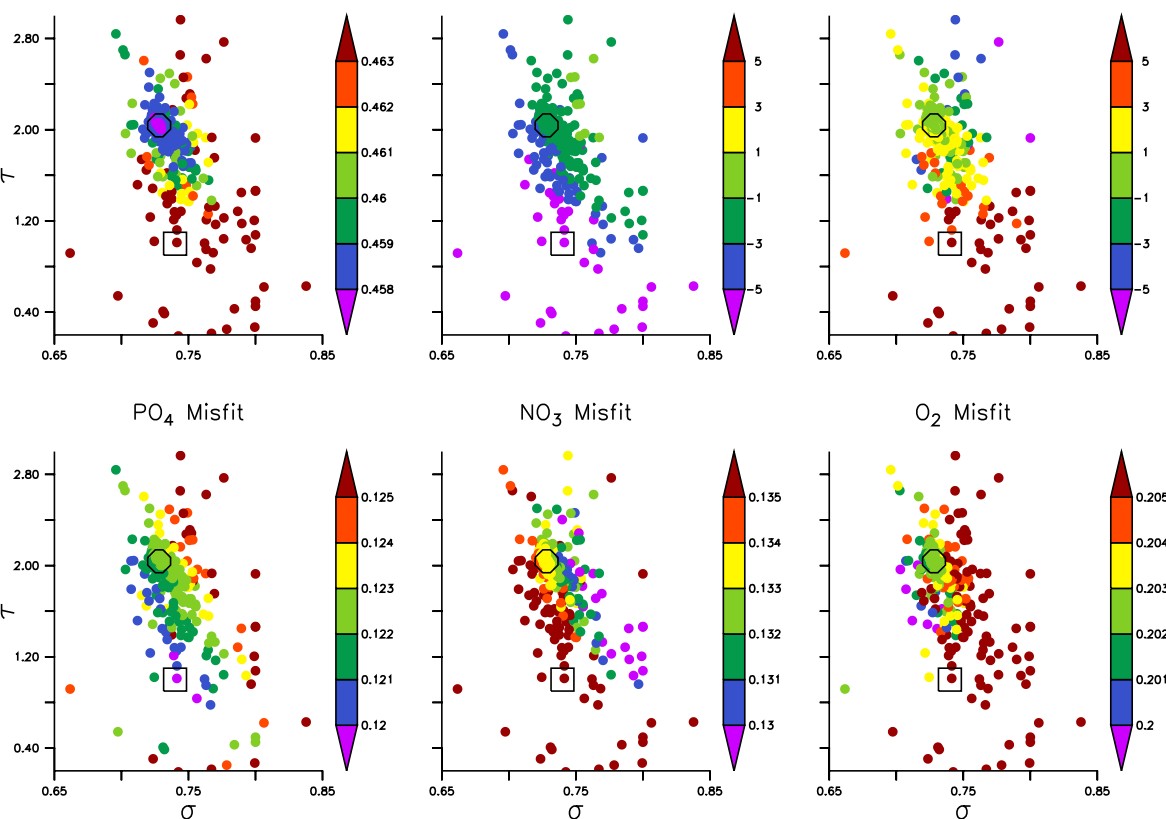

**Figure 8.** Model misfit and relative bias $b_j$ of RetroMOPS°, plotted for parameter combinations of $\sigma$ and DOP decay timescale $\tau$, where $\tau = 1/(\lambda + \lambda_s)$. Relative bias is evaluated by $b_j = (\overline{m_j}/\overline{o_j} - 1) \times 100$, where $\overline{m_j}$ denotes the global mean model concentration of tracer $j$, and $\overline{o_j}$ the observed mean. Model misfit is shown as total misfit ($J$ of Eqn. 11; upper left), and separated into it components, normalised by $\overline{o_j}$ ($J(j)$ of Eqn. 11; lower panels). The analysis is restricted to all individuals $i$ whose $b$ differs less than 5% from optimal $b^*$, i.e. $|b_i/b^* - 1| < 0.05$. For better visibility some model solutions ($\approx 10$), that are outside the range $0.65 \le \sigma \le 0.85$ and $0.2 \le \tau \le 3$ have been omitted from the plot. Open squares denote optimal estimates by Kwon and Primeau (2006, total phosphate constraint), open circles the optimal parameter from this study.