# Peer review of "Calibration of a simple and a complex model of global marine biogeochemistry"

_Biogeosciences, 2017_

## Referee Comment (RC1) · Anonymous Referee #1 · 5 Apr 2017

General comments

The author used a time-efficient technique to optimize biogeochemical model parameters from two global ocean biogeochemistry models of different complexities. A large uncertainty in climate modeling arises from highly parameterized representations of biogeochemical processes. Constraining uncertain model equations and their parameters has been a great challenge in climate modeling. Previous studies, including the ones led by the author, have addressed the challenge by developing an offline modeling approach in which equilibrium solutions of global ocean biogeochemistry models can be obtained in a time-efficient way. The offline model is combined with a time-efficient optimization technique to constrain biogeochemical model parameters of a NPZD type global ocean biogeochemistry model, which is mainly presented in Kriest et al., 2017. This study uses the same model and technique to constrain some of key parameters

controlling subsurface biogeochemical cycles. The author also takes a step forward by evaluating optimization results of two models with different complexity: a simple two-component model and the NPZD type seven-component model. I found the comparison interesting and informative, although the main conclusion from the comparison does not differ much from an earlier work by Kriest et al., 2010. Overall, the paper is well written, but I feel that the author needs to do more to highlight the novel aspects of this study.

Specific comments

The author chose to optimize 6 parameters from the NPZD type model and 4 parameters from the simpler model. The six parameters from the NPZD type model mostly represent remineralization processes of sinking organic particles, especially in suboxic conditions, whereas the four parameters from the simpler model represent remineralization of both dissolved organic matter and particulate organic matter. The choice of the parameter sets to be optimized vs to be retained from earlier studies seems arbitrary. Is there any particular reason or criteria by which the parameter sets are chosen for the optimization? The transport matrix method combined with the optimization technique (i.e., CMA-ES) seems a powerful way to constrain many uncertain biogeochemical model parameters. How would the results differ if the author optimized all of the parameters presented in Table 1 simultaneously? Remineralized nutrients are eventually transported to the euphotic layer and become available to support primary production. Therefore all parameters are interrelated with each other. In other words, optimal values for Ic, KPHY, $\mu$zoo, and $\kappa$zoo would be different with the newly optimized values for b*, KO2, KDIN, etc. This could affect the model evaluations as well.

Evaluation of DOP simulated from RetroMOPS remains qualitative. Although it is not sufficient, global datasets of DOM were presented in some previous studies including Letscher and Moore, 2015. How does the simulated DOP from the two models compare with the observations in terms of its distributions and concentrations? Why can't DOP be used as an extra constraint for the optimization in this study?

In many parts of the manuscript, statements are quite qualitative. For example, in conclusions, the author wrote "results in a slightly better fit to observed tracers, and in a much better fit to observed estimates of ..." The author should provide some quantitative assessment. How good is it compared to other experiments within this study and also compared with other studies?

Table 3: I don't understand what $r\Theta(\Omega)$ represents. How is a priori range determined? How should I interpret the values? Is it discussed in the text?

Fig. 6: It is interesting that the six panels are all different in terms of the pairs of the parameter values corresponding to the lowest misfit and bias. Does it mean that the optimal values for the two parameters (as presented in Table 3) are not optimal anymore if only subsets of the tracers are used as a constraint or if the bias is used as a cost function?

Technical corrections

Equation (2): This formulation does not look like the Martin curve. Is the term $z(j+1)$ right? Equation (11): Please check the last term Page #8, line #24: typo Page #12, line #1: There is no such a term like preformed waters. Perhaps change it to "reduces preformed DOP in subducted waters"?

---

## Referee Comment (RC2) · F. Hoffmann (Referee) · 10 Apr 2017

Please consider that I do not have a background in modelling, so I cannot judge the methods and main contents of the manuscript. With a background in functional microbiology of nitrogen cycling, my comments are limited to the representation of biological processes in the proposed model.

General comments: Key processes connected to organic matter remineralisation seem to be well represented also in the simplified model Retro-MOPS. The focus on the oxidants oxygen (for aerobic mineralisation) and nitrate (for anaerobic mineralisation) takes into account key remineralisation processes in the water column.

Specific comment: Page 5, line 10-11: "The discretised flux divergence, that can actually be remineralised to phosphate and nitrate with available oxidants (oxygen and/or

nitrate) Deff, (j), is then determined by...." - organic matter cannot be remineralized to nitrate by nitrate. In the absence of oxygen and the presence of nitrate, organic matter will be remineralized by denitrification (reducing nitrate to N2 while oxidizing the organic carbon), but the organic nitrogen in the organic matter will be remineralised and released as ammonium. This can in turn be oxidized to nitrate in the presence of oxygen (nitrification). Under low oxygen concentration, this process can occur parallel to denitrification (coupled nitrification-denitrification). But in the absence of oxygen, ammonium will not be oxidised to nitrate. I am not sure if this is important in this context, but the sentence as it stands now is misleading.

Technical correction: Page 3, line 21/22: remove one "the"

---

## Referee Comment (RC3) · Anonymous Referee #3 · 12 Apr 2017

**=Summary=**

The manuscript presents a parameter optimisation study involving a conventional marine biogeochemistry model, MOPS, and a low complexity model, RetroMOPS. Both models are optimised to full 3D, global fields of nutrients and oxygen, and both perform equally well after this process. The work finds that, as the processes that govern N and O2 distributions are decoupled from one another, both tracers represent valuable optimisation targets. In the case of the simple model, the work finds a relationship between DOP and phosphate, and suggests that observations of the former may help provide a useful model constraint.

=Review=

Overall, I found the manuscript an interesting, and relatively straightforward investigation into model calibration. Almost all of the issues that I have identified below with the manuscript are minor, mostly requests for additional clarification.

However, my major comment on the manuscript is that - considering its title - the manuscript is weak on the intercomparison of the simple and complex models. This is in contrast to a much stronger examination of the role of DOP in RetroMOPS.

Regarding the model intercomparison, in the first instance, this is hampered by limitations in the traceability between the models. For instance, both models have DOP (results of which are later shown to be interesting), but (a) MOPS remineralises this at a single rate while RetroMOPS has two rates for different depths, and (b) this rate is around double that in RetroMOPS compared to MOPS. Similarly, the models both differ in value for other shared parameters, and even whether parameters are optimised between the models. Given that there is a degree of overlap in the components of both models, I would expect, firstly, that these would be as aligned as much as possible, and, secondly, if unaligned, that it would be fully explained why not. A passing remark on parameters being "hand-tuned" is not enough. The result of all of this is that the intercomparison seems less systematic and more arbitrary than it otherwise could be. Additionally, the manuscript's discussion does not contextualise itself in a way that would help readers understand how what has been learned from this work could be applied to other models (the complex model here is still relatively simple, for instance).

Since the comparison between the two models is a major focus of the paper, at a minimum, I would expect a revised manuscript to better explain the seeming discrepancies in the choices made here, as well as more effort put into the intercomparison. Ideally, additional optimisations might be done to narrow the gaps between the models and permit a more complete intercomparison that would be of greater value to the marine biogeochemistry community.

My recommendation is publication after revision.

=Minor comments=
**==Abstract==**

Lines 6-7: The paper introduces two models of differing complexity, but beyond noting that the simpler model does almost as well as the complex one (which is not a given; e.g. Kwiatkowski et al., Biogeosciences, 2014), nothing else is said; more generally, this is more widely true of the manuscript – a better case / explanation should be made for presenting both the simple and complex models; especially as the two model structures are not traceably related to one another (though I'm sure that they share subroutines for specific processes)

==Introduction==

Lines 7-16: While I understand the need to keep manuscripts to a reasonable length, this paragraph is extremely dense and confusing; ideally, the concepts it describes should be clearly spelled out

Lines 7-10: In particular, this list of tools is mentioned in passing without any contextualising information; a sentence on each would be helpful to readers unfamiliar with them

Lines 17-18: This point about simulation time is slightly confusing here, since the preceding text is talking about accelerated models

Lines 27-31: As you latterly introduce oxidant dependent decay, it may be helpful to expand briefly on what parameters and processes Kriest et al. (2017) looked at

Line 31: This sentence could do with an example, for instance "(e.g. parameter X was found to play a role in vertical distributions of process Y)"

Pg. 3, line 2: "have been popular" - presumably this refers to use in CMIP, etc.?

==Methods==

Pg. 3, line 28: You say "A fraction" but don't give a value; other parameter values are listed; what's this one?

BGD
Pg. 3, line 30: "Non-buried detritus is resuspended into the water column" – this sounds intriguing, care to expand? For instance, resuspension over how much of the water column? What about the benthic boundary layer?

Pg. 3-4: Simple flow schematics of the two models would probably be helpful

Pg. 4, lines 19-20: It is not immediately clear to me that the absence of the explicit plankton reservoirs in MOPS means that remineralisation would be too slow in Retro-MOPS; arguably, the storage of material within particulate reservoirs for a period of time might instead act to slow down remineralisation back to useable nutrient; in fact, is that not what Table 3 reports for its optimised value for this parameter?

Pg. 5, equation 4: Why not just expand on what s\_O2(j) and s\_DIN(j) are here instead of sending the reader to another manuscript?

Pg. 5, equations 4-5: These seem to imply that you calculation potential remineralisation, then calculate the possible remineralisation given O2 and DIN, then apply the latter at level k but apply the remainder at level k+1; how does this deal with the situation where level k+1 has insufficient oxidant?; that doesn't seem all that unlikely in OMZs (though with low vertical resolution as here, this may be less of a concern); more obviously, why mess around with where the remineralisation flux occurs and not just stop the sinking flux from remineralising?; for instance, couldn't the calculation of D(j) not use O2 and DIN to affect how the export flux is attenuated?; i.e. when there's no oxidant, remineralisation cannot occur vs. remineralisation occurs, where the required oxidant is taken from is dealt with afterwards

Pg. 6, line 2: In saying "nitrogen fixation balances the simulated loss", this implies a direct connection which does not appear to be the case in equation 6; instead, the model losses and gains inevitably come into a balance, but they are not directly linked (some other models do make this connection)

Pg. 6, equations 7-10: With unwieldy equations like these, underbraces can be helpful
in providing a quick reference for the reader as to the identity of the terms

Pg. 6, section 2.3: This overlooks any statement as to the performance of the physical model, even one that simply cites a source on this; given that the whole ocean misfit is used as an optimisation target, letting readers know that there's not a strong ventilation bias in the model ocean might be useful; this has a relationship with the next point ...

Pg. 6, line 26: Is the Marshall et al. (1997) reference is the source of the circulation state used here?; or is it based on a more recent simulation?

Pg. 7, line 1: "After 3000 years most tracers have approached steady state" – This is an oddly loose definition of equilibrium; you could instead refer to the stabilisation of misfit J (e.g. that it fixes to N decimal places)

Pg. 7, lines 24-25: While the normalisation to global concentration should help with N and P (since they are related linearly through the ocean), does this overplay or underplay O2?; this doesn't show the same sort of relationships (for obvious reasons); also, you don't mention AOU at all – would this be a good alternative misfit target?

Pg. 8, line 15: I might not have waited until the last sentence of this section to explain about the default parameter set

Pg. 8, section 2.7: Why is R\_-O2:P optimised in MOPS but not in RetroMOPS?; I can't immediately see why this isn't an option

Pg. 8, section 2.7: MOPS is optimised with a reduced data set but RetroMOPS is not; this seems like a strange omission considering the same underlying issue affects both models; it is again symptomatic of the disparities between the models being intercompared

Pg. 8, lines 26-28: Is the numerical diffusion referred to here what's decribed in Section 3.3?; might be worth teeing the reader up for this; also does horizontal diffusivity play a role at all?; the model is pretty coarse in the horizontal as well (though the gradients may be weaker)

BGD
==Results and Discussion==

Pg. 9, line 10: Missing "of", i.e. "Because of optimisation, MOPSoD results ...."

Pg. 9, line 17: Typo on "threshold"

Pg. 9, line 20: The statement "impose a threshold" is unclear; do you mean that denitrification could not occur below this concentration?

Pg. 9, lines 25-34: What does this omitting of the Equatorial Pacific to the misfit in this region?; is it better or worse than when it's included in the global misfit function?

Pg. 9, lines 25-34: Also, what about the reverse situation where only the Equatorial Pacific (and / or OMZs more generally) is used for tuning?; if you tried that, perhaps a passing remark on it would be interesting

Pg. 10, lines 8-17: You note in the manuscript that "nitrogen fixation counteracts denitrification" but, as mentioned above, there's no direct connection in the model (e.g. unlike some models that represent the former implicitly as a function of the explicit latter); in the context of (dis-)equilibrium, I don't have a feel for the relative rates of the two processes in the work here; I guess I'm wondering if certain combinations of parameter values promote or diminish equilibriation time; I suspect this is unlikely, but optimisation can take models to strange places

Pg. 10, lines 20-22: Per previous remarks on circulation, how good is the ocean's ventilation?; reporting CFC or (especially) C-14 performance earlier would help (even if this reiterates previous work)

Pg. 10, line 23: One of the lambdas in the bracketed comment should be the surface DOP remineralisation rate; also, only one of them is given units

Pg. 11, line 15: Does "direct evaluation of steady state" mean that they calculated the steady state solution analytically?

Pg. 11, lines 15-16: "... may still exhibit some drift ..."?; it would probably be helpful
to make this clearer, or possibly quantify it (e.g. in terms of misfit J fit; see my earlier remark); by all means use a fixed simulation duration, but knowing what this means for the misfit measure would be useful (e.g. its drift rate at this point)

Pg. 12, line 4: separate this last part of Section 3.3 into Section 3.4?; as it's on the comparison of MOPS and RetroMOPS, it would make a clear section; it might also afford an location to delve a little deeper into the complexity issue that's currently rather glossed over in the draft manuscript

Pg. 12, line 27: remove spare comma to get "... (Table 4), and indicates that these tracers ..."

==Conclusions==

Pg. 13, line 12: Regarding the use of observational DOP, can you clarify somewhere in the text how homogeneous DOP is?; i.e. is a single remineralisation timescale likely to be representative?

==Acknowledgements==

Pg. 13: Is this paper part of a special issue or wider celebration of the life of Ernst Maier-Reimer?; if so, an earlier note in the introduction would seem to be in order; if not, it may be worth making the rationale for this tribute a little clearer (e.g. note Maier-Reimer's recent passing)

==Figures and Tables==

Table 1: Why is the ostensibly fixed parameter DINmin very slightly different in the two RetroMOPS runs?

---

## Referee Comment (RC4) · Anonymous Referee #4 · 24 Apr 2017

**General comments**

The author presents results from optimisation experiments with two relatively simple global biogeochemistry models that are notable for their representation of oxidant-limited remineralisation. The second model is a simplified version of the first, but performs similarly well to the first with regard to reproducing global PO4, NO3 and O2 concentrations as the model approaches a steady state. This finding is the main focus of the paper in the title and abstract.

Despite this focus, much of the paper is given over to comparing a previous optimisation of the more complex model to surface observations, to a new optimisation to deep observations. The author finds that parameters relating to remineralisation, stoichiometry and the oxidant limitation of remineralisation are well constrained by deep-ocean

observations of  $PO_4$ ,  $NO_3$  and  $O_2$ .

My overall feeling is that the paper is badly lacking in focus. Reading through I was always struggling to understand what major point the author was hoping to make. Is it that the simple model is nearly as good as the complex model, or is it that different parts of the model are better constrained by different kinds of observations? At the moment the article reads as if two separate (and somewhat poorly developed) stories have been combined into one, with very little thought as to what connects them. I think that the author either needs to pick one theme, and develop it better, or needs to do a much better job of finding a narrative thread linking the two themes together. It is up to the author to identify how that might be achieved.

Specific comments

Abstract

Line 7: "a complex seven-component model (MOPS), and a very simple twocomponent model (RetroMOPS)" and "The simple model, which contains only nutrients and dissolved organic phosphorus (DOP)". RetroMOPS clearly has four components:  $PO_4$ ,  $NO_3$ ,  $O_2$  and POM.

Line 13: Please do a better job of explaining what is "the global bias".

1 Introduction

Line 29: "[Kriest et al. (2017)] showed that annual mean tracer concentrations do not provide much information on parameters related to the dynamic biological processes taking place in the euphotic zone". Should be "annual mean tracer concentrations did not provide much information", as I am not convinced this is a general result for all models.

2.2.1 Primary production

Equation 1: Why use the mean phytoplankton concentration at all? It would be more
consistent with the rest of the model (i.e. Equation 5) to convolve the specific growth rate and the phytoplankton concentration into a single growth rate of the phytoplankton population (mmol P m-3 d-1).

2.2.2 The fate of primary production: Export, DOP production and remineralisation

Line 19: "DOP then decays to phosphate and nitrate". To me it would make sense to call it POM.

Line 19: "To allow for a potential, fast recycling loop at the surface, RetroMOPS parameterises an additional decay rate". Presumably this is inspired by (Oschlies 2001), but why would this be necessary in the absence of assimilated primary production observations?

Equation 4: I think a bit more could be said about the interdependence of  $s_{O2(j)}$  and  $s_{DIN(j)}$ . For example, their sum forms a coefficient for remineralisation, so it is important to note that their sum is constrained between 0 and 1.

**2.5 Misfit function**

Equation 11: I am a bit confused by how the misfit function and its components are defined. In particular, I cannot see how  $\bar{o}_j$  (the global average observed concentration of the respective tracer) is included in the RHS.

Also, it seems that the model is being compared to gridded observations, instead of observational equivalents being extracted at the spatiotemporal locations of the observations. As the gridding process will introduce its own set of errors, this choice needs some justification.

2.6 Optimisation of MOPS

Line 15: I don't think including results from the hand-tuned model brings anything of value to the paper.

3.1 Optimisation of MOPS

BGD
Line 4: Fig 1 (rather than Fig S1)?

Line 4: 10% and 1% (rather than 10% and  $1^{\circ}/_{\circ\circ})?$

Line 12 (and throughout the text): "reduced denitrification". It is probably safer to avoid the word reduced except with regard to the chemical process.

3.3 Optimisation of RetroMOPS

Line 21: "The misfit to phosphate (Fig. 8, lower left panel) shows an elongated valley in the two-dimensional projection". I do not see a valley in this Figure. The misfit slopes down from the top-right towards the bottom left, but there is nothing to suggest it slopes back up again after reaching a minimum.

References

Oschlies, A. (2001) Model-derived estimates of new production: New results point to lower values. Deep-Sea Res. II. 48, 2173–2197.

---

## Author Response (AR1)

I thank the four referees for their detailed comments. Below is my point-by-point response to them (as already published on BGs discussion website). This is followed by a marked-up version of the revised manuscript, that includes all the according changes I made, plus some minor changes with respect to wording. In the revised version I also discuss the recent paper by Galbraith et al. (2015) who compared the outcome of three different biogeochemical models, simulated in a common earth system model. This paper just recently came my attention.

Reply to referee 1:

I thank referee 1 for his/her constructive and helpful comments. Below are my answers to the comments, indicated by "IK:"

"I found the comparison interesting and informative, although the main conclusion from the comparison does not differ much from an earlier work by Kriest et al., 2010."

IK: Indeed the results obtained in this study point into the same direction as those of Kriest et al. (2010), the exception being that here via optimisation I use a more thorough and objective scan of the parameter space, with many different combinations of parameters. I will try to point this out in a revised version of the paper.

"The choice of the parameter sets to be optimised vs to be retained from earlier studiesseems arbitrary. Is there any particular reason or criteria by which the parameter sets are chosen for the optimization?"

IK Yes, there is - I am sorry that this did not come across properly: The first optimisation (MOPSoS; Kriest et al., 2017) was basically meant as a test for the framework, and a proof-of-concept. I therefore chose a very wide range of parameter types, across all trophic levels, and acting on different time and space scales (see also Kriest et al., 2012, for the large scale effect of "b"). As it turned out, mainly the surface parameters couldn't be constrained by the present misfit function; so I exchanged those by parameters related to deeper processe (MOPSoD). The selection of those was motivated by the large uncertainty regarding extent and expansion of oxygen minimum zones in models (Cocco et al., 2013; Cabre et al, 2013), and by the fact, that very little knowledge exists about these parameters (or even parameterisations). As it turned out, some of these were quite difficult to constrain, probably owing to the small volume of OMZs, relative to total ocean volume. Given that in RetroMOPS especially the parameters for DOP production and decay have to act as a shortcut for the entire complex surface ecosystem, I chose to optimise those, in addition to "b" and the -02:P stoichiometry (as control). I will try to make this approach clearer in a revised version of the paper.

"How would the results differ if the author optimized all of the parameters presented in Table 1 simultaneously? Remineralized nutrients are eventually transported to the euphotic layer and become available to support primary production. Therefore all parameters are interrelated with each other. In other words, optimal values for Ic, KPHY,  $\mu$ zoo, and Kzoo would be different with the newly optimized values for b\*, KO2, KDIN, etc. This could affect the model evaluations as well."

IK: I agree that it would be most desirable to optimise all parameters at once. However, given what was found by Ward et al. (2010), and the possible interdependence of some of these parameters, this did not seem feasible to me. I will explain the choice of parameters in a revised version of the manuscript.

"Evaluation of DOP simulated from RetroMOPS remains qualitative. Although it is not sufficient, global datasets of DOM were presented in some previous studies including Letscher and Moore, 2015. How does the simulated DOP from the two models compare with the observations in terms of its distributions and concentrations? Why can't DOP be used as an extra constraint for the optimization in this study?"

IK: It can, and should, to my opinion, be part of the misfit function. However, as the global DOP data distribution is very sparse (compared to nutrients and oxygen), this may require a special treatment or weighting in the misfit function. We are currently working on different ways and methods to account for DOP in model calibration. A thorough treatment and discussion of this would, I fear, exceed the scope of this paper, which is already quite diverse.

"In many parts of the manuscript, statements are quite qualitative. For example, in conclusions, the author wrote "results in a slightly better fit to observed tracers, and in a much better fit to observed estimates of ..." The author should provide some quantitative assessment. How good is it compared to other experiments within this study and also compared with other studies?"

IK: I will try to be more specific in the conclusion section of the revised version of this paper.

"Table 3: I don't understand what  $r\theta(\Omega)$  represents. How is a priori range determined? How should I interpret the values? Is it discussed in the text?"

IK: This is the range of of parameter values of "very good" individuals, i.e. those model solutions, whose misfit is not worse than 1 permil of the best individual, divided by the "allowed" parameter range. For those parameters with a very wide range (or a priori uncertainty), the chance of having a huge spread in final solutions might just be larger - this is why I showed this normalised value of  $R\Theta(\Omega)$ . I will try to explain this better in a revised version of the manuscript.

"Fig. 6: It is interesting that the six panels are all different in terms of the pairs of the parameter values corresponding to the lowest misfit and bias. Does it mean that the optimal values for the two parameters (as presented in Table 3) are not optimal anymore if only subsets of the tracers are used as a constraint or if the bias is used as a cost function?"

IK: Yes: e.g., if we chose only oxygen RMSE for the misfit function, we would probably end up with a different best set of parameters than when only using nitrate.

Technical corrections

"Equation (2): This formulation does not look like the Martin curve. Is the term z(j+1) right?"

IK: No, it wasn't. It should indeed be z(k+1). Thank you for spotting this! "Equation (11): Please check the last term"

IK: The  $1/bar(o_j)$  was in the wrong place: it should have been after the first sum of the second term. Thank you for drawing my attention to this.

"Page #8, line #24: typo" IK: Will be corrected.

"Page #12, line #1: There is no such a term like preformed waters. Perhaps change it to "reduces preformed DOP in subducted waters"?"

IK: Yes, thank you!

Reply to referee 2:

"Specific comment: Page 5, line 10-11: "The discretised flux divergence, that can actually be remineralised to phosphate and nitrate with available oxidants (oxygen and/or nitrate) Deff, (j), is then determined by...." - organic matter cannot be remineralized to nitrate by nitrate. In the absence of oxygen and the presence of nitrate, organic matter will be remineralized by denitrification (reducing nitrate to N2 while oxidizing the organic carbon), but the organic nitrogen in the organic matter will be remineralised and released as ammonium. This can in turn be oxidized to nitrate in the presence of oxygen (nitrification). Under low oxygen concentration, this process can occur parallel to denitrification (coupled nitrification-denitrification). But in the absence of oxygen, ammonium will not be oxidised to nitrate. I am not sure if this is important in this context, but the sentence as it stands now is misleading."

IK: I thank Friederike Hoffmann for drawing my attention to this quite erroneous and misleading expression in the paper. Indeed, in MOPS organic matter is denitrified to phosphate and N2 (without any intermediate reduced N compounds). To avoid any confusion, I will skip to "phosphate and nitrate"

Reply to referee 3:
* * *
I thank referee 3 for his/her thorough reading and instructive comments. Below are my answers to the comments, indicated by "IK:"

"Regarding the model intercomparison, in the first instance, this is hampered by limitations in the traceability between the models. For instance, both models have DOP (results of which are later shown to be interesting), but (a) MOPS remineralises this at a single rate while RetroMOPS has two rates for different depths, and (b) this rate is around double that in RetroMOPS compared to MOPS. Similarly, the models both differ in value for other shared parameters, and even whether parameters are optimised between the models. Given that there is a degree of overlap in the components of both models, I would expect, firstly, that these would be as aligned as much as possible, and, secondly, if unaligned, that it would be fully explained why not. A passing remark on parameters being "hand-tuned" is not enough. The result of all of this is that the intercomparison seems less systematic and more arbitrary than it otherwise could be. Additionally, the manuscript's discussion does not contextualise itself in a way that would help readers understand how what has been learned from this work could be applied to other models (the complex model here is still relatively simple, for instance).

IK: I will try to present the approach I followed in a more concise way, and will discuss the finding (Given this particular misfit function, a model that shortcuts biology at the surface van perform almost as good as a model with more detail) before the background of Kwiatkowski et al. (2014; see also below). I will further slightly restructure the paper to better emphasize the main points.

"Since the comparison between the two models is a major focus of the paper, at a minimum, I would expect a revised manuscript to better explain the seeming discrepancies in the choices made here, as well as more effort put into the intercomparison. Ideally, additional optimisations might be done to narrow the gaps between the models and permit a more complete intercomparison that would be of greater value to the marine biogeochemistry community."

IK: I am sorry that the degree of alignment between MOPS and RetroMOPS did not show up clearly enough: e.g., both models are exactly the same with respect to nutrient and light sensitivity of primary production. Also, the decay rate constants and oxidant sensitivities are both the same. The different parameterisations of DOP production and decay arises because this component in RetroMOPS acts as a kind of "shortcut" of the grazing-remineralization cycle in MOPS. In fact, the two different degradation rate constants were introduced to allow an even greater similarity between MOPS and RetroMOPS, allowing RetroMOPS a more flexible mimicking of a potential fast nutrient turnover. I will try to explain this better, and discuss potential insights and consequences in more detail, in a revised version of the manuscript. Given that circulation, and also the formulation of the misfit function are most probably two other, highly important factors in biogeochemical model calibration, and that optimisation is a computing-time consuming issue, I would nevertheless prefer to keep this set of experiments presented here, and in future runs concentrate on physical and observational errors and choices.

**==Abstract==**

"Lines 6-7: The paper introduces two models of differing complexity, but beyond noting that the simpler model does almost as well as the complex

one (which is not a given; e.g. Kwiatkowski et al., Biogeosciences, 2014), nothing else is said; more generally, this is more widely true of the manuscript — a better case / explanation should be made for presenting both the simple and complex models; especially as the two model structures are not traceably related to one another (though I'm sure that they share subroutines for specific processes)"

IK: This comment by the reviewer addresses two points: (a) the (lack of) introduction of RetroMOPS, as well as (b) its performance with respect to the skill metrics to biogeochemical tracers.

(a) When developing RetroMOPS I tried to include as many features of previous simple models global biogeochemical models (e.g., Bacastow and Maier-Reimer, 1991; Kriest et al., 2010, 2012), while at the same time maintaining MOPS's remineralisation scheme. The resulting "compromise" RetroMOPS is therefore not directly traceable to MOPS; however, the more gradual transition between model complexity (N, N+DOP, NP+DOP, NPZ+DOP, NPZD+DOP) presented in Kriest et al. (2010, 2012) might, to some extent, provide some more insight into effects of adding complexity on model skill. The revised version will include some more discussion on the current results before the background of this paper.

(b) I have tried to point out in the paper that it depends on the research question to be addressed with a model, which metric to choose; this will in turn determine which model is best suited to address that question. So far, the optimizations presented here indicate that a more complex model does not necessarily outperfom a simpler one. I will try to emphasize this even more in a revised version of the manuscript. However, I do not think that the results obtained by Kwiatkoski et al. (2014) contradict this finding, and suggest to put my results into a wider context by referring to their work: Kwiatkowski et al. (2014) compared six different global biogeochemical models, coupled to NEMO (1x1 spatial resolution and 75 vertical levels), and simulated over 118 years, against data sets of surface pCO2, DIC, alkalinity, DIN, Chl a and primary production. The models of vary in complexity from seven to 57 compartments, and thus also in their computational demand by almost a factor of five.

To assess model skill they ranked the models with respect to spatial correlation between and variance of model and observations. In general, the more complex models perform better with respect to simulated variance, but the simpler models better with respect to spatial correlation. Although no model outperforms all models across all metrics, they conclude that "Results suggest that little evidence that higher biological complexity implies better model performance in reproducing observed global-scale bulk properties."

This conclusion may be even more obvious when taking into account the ability of the different diagnostics to distinguish among the models: For example, spatial correlation of DIN ranges only betwee 0.79 to 0.94. Even more, for DIN, alkalinity and DIC normalized standard deviations vary less than 10% around the average standard deviation. Excluding these diagnostics from the model assessement would result in an advantage for the simpler models (MEDUSA or HADOCC) with respect to spatial correlation and a quite good performance of these model with respect to standard deviation (sum of ranks 10 and 9 for MEDUSA and HADOCC, respectively, compared to seven and 10 for PlankTOM6 and PlankTOM10). Finally, some of the models differ only very slightly in their performance (e.g., a difference between r=0.93 and r=0.92 for spatial correlation of DIC), in my opinion hampering the applicability of ranking.

Although it is clear that intermediate complexity models such as HADOCC cannot represent the level of detail embedded in more complex models, and that it cannot be ruled out that "more complex models can in future provide additional insight based on ongoing measurements and data syntheses", so far the model evaluation with respect to the bulk, biogeochemical observations such as dissolved inorganic tracers or chlorophyll does not seem to indicate any superiority of more complex models on a global scale. Although Kwiatkowski et al. (2014) apply very different temporal and spatial scales (given by the much shorter model spinup and focus on surface diagnostics), the results obtained with RetroMOPS and MOPS corroborate their findings. As noted by them, future availability of more complex data sets, such as different plankton groups, or particle distributions, will provide further insight about

the level of model complexity required, given the research question to be addressed with a model.

==Introduction==

"Lines 7-16: While I understand the need to keep manuscripts to a reasonable length, this paragraph is extremely dense and confusing; ideally, the concepts it describes should be clearly spelled out"

IK: I agree, and will add some sentences on the different methods applied.

"Lines 7-10: In particular, this list of tools is mentioned in passing without any contextualising information; a sentence on each would be helpful to readers unfamiliar with them"

IK: I agree, and will add some sentences on the different methods applied.

"Lines 17-18: This point about simulation time is slightly confusing here, since the preceding text is talking about accelerated models."

IK: Indeed, it is confusing and misleading. I will skip the reference to simulation time.

"Lines 27-31: As you latterly introduce oxidant dependent decay, it may be helpful to expand briefly on what parameters and processes Kriest et al. (2017) looked at"

IK: I will add some explanation on decay parameters and processes.

"Line 31: This sentence could do with an example, for instance "(e.g. parameter X was found to play a role in vertical distributions of process Y)""

IK: I agree, and will add an example of the effect of b on large spatial phosphate distribution, with reference to Kriest et al. (2012)

"Pg. 3, line 2: "have been popular" - presumably this refers to use in CMIP, etc.?"

IK: Not necessarily to CMIP (complexity of many current CMIP models is more similar to MOPS), but to models used by e.g., Bacastow and Maier-Reimer, 1991; Matear and Hirst, 2003; Kwon et al., 2006; Dutkiewicz et al., 2006-

==Methods==

"Pg. 3, line 28: You say "A fraction" but don't give a value; other parameter values are listed; what's this one?"

IK: The fraction buried depends on the deposition rate onto the sediment (Kriest and Oschlies, 2013); I will add a brief description on it in the revised version of the paper.

"Pg. 3, line 30: "Non-buried detritus is resuspended into the water column" — this sounds intriguing, care to expand? For instance, resuspension over how much of the water column? What about the benthic boundary layer?"

IK: In fact, it is resuspended evenly in the last bottom box (i.e., there is no BBL). Effects of this have been investigated extensively in Kriest and Oschlies, 2013.

"Pg. 3-4: Simple flow schematics of the two models would probably be helpful"  $% \left[ \left( {{{\mathbf{r}}_{\mathbf{r}}}_{\mathbf{r}} \right)^{2}} \right] \right]$

IK: I will add two flow charts of MOPS and RetroMOPS.

"Pg. 4, lines 19-20: It is not immediately clear to me that the absence of the explicit plankton reservoirs in MOPS means that remineralisation would be too slow in RetroMOPS; arguably, the storage of material within particulate reservoirs for a period oftime might instead act to slow down remineralisation back to useable nutrient; in fact, is that not what Table 3 reports for its optimised value for this parameter?" IK: Yes, indeed - thank you for the comment. I will add some discussion on this in the revision. See below, your comments and my reply re. Conclusions section: Given the rather fast turnover rates of DOP observed by Hopkinson et al. (2002), I did not find it appropriate to have a specific slowdown of remineralization.

"Pg. 5, equation 4: Why not just expand on what s\_O2(j) and s\_DIN(j) are here instead of sending the reader to another manuscript?"

IK: I will add some more description on this in the revised version (see also my response to Lines 27-31).

"Pg. 5, equations 4-5: These seem to imply that you calculation potential remineralisation, then calculate the possible remineralisation given O2 and DIN, then apply the latter at level k but apply the remainder at level k+1; how does this deal with the situation where level k+1 has insufficient oxidant?; that doesn't seem all that unlikely in OMZs (though with low vertical resolution as here, this may be less of a concern); more obviously, why mess around with where the remineralisation flux occurs and not just stop the sinking flux from remineralising?; for instance, couldn't the calculation of D(j) not use O2 and DIN to affect how the export flux is attenuated?; i.e. when there's no oxidant, remineralisation cannot occur vs. remineralisation occurs, where the required oxidant is taken from is dealt with afterwards"

IK: In the case layer k+1 also has insufficient oxidants, the organic matter will propagate further downwards, until it reaches sufficient oxidants, or the sea floor to be buried. Using this scheme I tried to be as close as possible to MOPS, where the explicit detritus sinks with its prescribed sinking speed, but only remineralises when it encounters enough oxygen or nitrate. So I think I have parameterised the model as suggested by reviewer 3; however, Eqn. 3 first presents the more general case (without oxidant dependency), which has been widely used in former simple global models.

"Pg. 6, line 2: In saying "nitrogen fixation balances the simulated loss", this implies a direct connection which does not appear to be the case in equation 6; instead, the model losses and gains inevitably come into a balance, but they are not directly linked (some other models do make this connection)"

IK: In the model nitrogen fixation balances denitrification on large time and space scales. It depends on biogeochemical parameters and circulation, how fast the two processes are connected (see also Kriest and Oschlies, 2015). I will add "in the long term" in a revised version of the manuscript, and add some words on the potential spatial distinction.

"Pg. 6, equations 7-10: With unwieldy equations like these, underbraces can be helpful in providing a quick reference for the reader as to the identity of the terms"

IK: This is a very good suggestion, thank you!

"Pg. 6, section 2.3: This overlooks any statement as to the performance of the physical model, even one that simply cites a source on this; given that the whole ocean misfit is used as an optimisation target, letting readers know that there's not a strong ventilation bias in the model ocean might be useful; this has a relationship with the next point ... Pg. 6, line 26: Is the Marshall et al. (1997) reference is the source of the circulation state used here?; or is it based on a more recent simulation?"

IK: This is the source of the circulation - see also Khatiwala et al. (2004) and Khatiwala (2008). The TM was derived from a 2.8×2.8 global configuration of the MIT model with 15 vertical layers, forced with monthly mean climatological fluxes of heat and freshwater, and subject to a weak restoring of surface temperature and salinity to observations. The circulation is detailed in Dutkiewicz et al (2005) and its configuration is similar to that applied in the Ocean Carbon-cycle Model Intercomparison Studies (OCMIP) (Orr et al, 2002). Circulation has been assessed within the OCMIP-2 project against a series of diagnostics and observations, such as T, S, and MLD (Doney et al. 2004), CFCs (Dutay et al. 2002; Matsumoto et al. 2004) and radiocarbon (Matsumoto et al.2004; also Graven et al. 2012). These studies suggest a good overal perfomance comparable to other models, with some weaknesses (too much North Pacific intermediate waters, AABW water formation only in Drake passage; unrealistic spreading of the CFC-11 signal into the interior of the deep ocean in the deep western boundary current of the Atlantic), and strengths (e.g., mode water formation in the Antarctic). Depending on diagnostic applied, waters may appear too young in that model, although this is influenced by the upper boundary condition of the respective age tracer (Koeve et al., 2012). I will add some sentences on this in the discussion, but would prefer to direct the interested reader directly to these papers.

"Pg. 7, line 1: "After 3000 years most tracers have approached steady state" — This is an oddly loose definition of equilibrium; you could instead refer to the stabilisation of misfit J (e.g. that it fixes to N decimal places)"

IK: I have not checked for the stabilisation of the misfit for all model simulations, but I agree, this is a very useful information to have. In fact, J for the two optimal runs is stable (at least up to e-4) after 3000 years. I will add a plot of the transient misfit function of these runs to the supplement, and note, that this might - to some extent - depend on the parameter settings (cf Kriest and Oschlies, 2015).

"Pg. 7, lines 24-25: While the normalisation to global concentration should help with N and P (since they are related linearly through the ocean), does this overplay or underplay O2?; this doesn't show the same sort of relationships (for obvious reasons); also, you don't mention AOU at all - would this be a good alternative misfit target?"

IK: N and P in this model are not linearly related throughout the ocean, because P is conserved (and only affected by "Redfieldian" processes), while N (either nitrate or fixed nitrogen) may change due to nitrogen fixation and denitrification. Both contribute to 20%-30% to the misfit function, while oxygen contributes about 40-50% (see also Kriest et al., 2017, Figures 4, 10, 13). With regard to "overplay" or "underplay" of oxygen, I think it depends on what we are interested in: if OMZs, we might even wantto stress this tracer in the misfit function. Yes, AUO (or EOU) could be a very useful diagnostic for the misfit function; likewise preformed nutrients.

"Pg. 8, line 15: I might not have waited until the last sentence of this section to explain about the default parameter set"

IK: I will move this to the beginning of the paragraph.

"Pg. 8, section 2.7: Why is R\_-O2:P optimised in MOPS but not in RetroMOPS?; I can't immediately see why this isn't an option"

IK: In MOPS' optimisations,  $R_-O2:P$  did not show any significant deviations from its default value of 170, so I decided to keep this fixed, and only vary those parameters, that relate to the parts of RetroMOPS that are very different to MOPS (see above). I will try to explain this better in the revision.

"Pg. 8, section 2.7: MOPS is optimised with a reduced data set but RetroMOPS is not; this seems like a strange omission considering the same underlying issue affects both models; it is again symptomatic of the disparities between the models being intercompared"

IK: Because the reduced data set did not show any large changes in estimated parameters or misfit of MOPS, and because I assume that other issues might be more important to investigate in an optimisation context (circulation; components and form of the misfit function), I decided not to spend computational resources on RetroMOPS with reduced data set. This "reduced data" experiment was mainly aimed at investigating the potential effect of circulation in the equatorial Pacific on the parameter estimate. Possibly due to its small area, the effect is almost negligible.

==Results and Discussion==

"Pg. 9, line 10: Missing "of", i.e. "Because of optimisation, MOPS^OD results . . . "" IK: Will be corrected.

"Pg. 9, line 17: Typo on "threshold"" IK: Will be corrected.

"Pg. 9, line 20: The statement "impose a threshold" is unclear; do you mean that denitrification could not occur below this concentration?"

IK: Yes. I will rephrase this.

"Pg. 9, lines 25-34: What does this omitting of the Equatorial Pacific to the misfit in this region?; is it better or worse than when it's included in the global misfit function?"

IK: The misfit for the equatorial Pacific becomes slightly worse (by about 3%) when omitting this region from the misfit function. I will add a few sentences on this in the revision.

"Pg. 9, lines 25-34: Also, what about the reverse situation where only the Equatorial Pacific (and / or OMZs more generally) is used for tuning?; if you tried that, perhaps a passing remark on it would be interesting"

IK: No, I did not try this as I suspect it is to a large extent the physics (e.g., missing equatorial jets) that causes the BGC misfit here.

"Pg. 10, lines 8-17: You note in the manuscript that "nitrogen fixation counteracts denitrification" but, as mentioned above, there's no direct connection in the model (e.g. unlike some models that represent the former implicitly as a function of the explicit latter); in the context of (dis-)equilibrium, I don't have a feel for the relative rates of the two processes in the work here; I guess I'm wondering if certain combinations of parameter values promote or diminish equilibration time; I suspect this is unlikely, but optimisation can take models to strange places"

IK: As we have shown (Kriest et al., 2015) sinking speed ("b") is one parameter that connects regions of N-loss with regions of N-gain (of course, before the background of circulation); DOM and POM and its lability will most likely be another candidate.

However, by year 3000 many of the models should have approached (more or less) equilibrium (Kriest and Oschlies, 2015, Figures 2 and 3).

"Pg. 10, lines 20-22: Per previous remarks on circulation, how good is the ocean's ventilation?; reporting CFC or (especially) C-14 performance earlier would help (even if this reiterates previous work)"

IK: See my answer above; I will add some sentences on this in the revision, but would like to keep my focus on the biogeochemistry in the current paper.

"Pg. 10, line 23: One of the lambdas in the bracketed comment should be the surface DOP remineralisation rate; also, only one of them is given units"

IK: Yes, the second one. I will correct this and add the unit.

"Pg. 11, line 15: Does "direct evaluation of steady state" mean that they calculated the steady state solution analytically?"

IK: No, not analytically, but using the Newton scheme involving the model's Jacobian. I will rephrase this in the revision of the manuscript.

"Pg. 11, lines 15-16: ". . . may still exhibit some drift . . ."?; it would probably be helpful to make this clearer, or possibly quantify it (e.g. in terms of misfit J fit; see my earlier remark); by all means use a fixed simulation duration, but knowing what this means for the misfit measure would be useful (e.g. its drift rate at this point)"

IK: See above: I will add figures on the transient of the misfit function to the supplement.

"Pg. 12, line 4: separate this last part of Section 3.3 into Section 3.4?; as it's on the comparison of MOPS and RetroMOPS, it would make a clear section; it might also afford an location to delve a little deeper

into the complexity issue that's currently rather glossed over in the draft manuscript"

IK: Yes, thank you very much for this suggestion - I will do that.

"Pg. 12, line 27: remove spare comma to get ". . . (Table 4), and indicates that these tracers . . . ""

IK: Will be corrected.

**==Conclusions==**

"Pg. 13, line 12: Regarding the use of observational DOP, can you clarify somewhere in the text how homogeneous DOP is?; i.e. is a single remineralisation timescale likely to be representative?"

IK: Hopkinson et al. (2002) applied a multi-G model to incubations of DOP sampled in surface waters of the middle Atlantic Bight, and measured decay constants for the very labile fraction of 0.22 per day (79 and 29-252 per year), with a range of 0.08-0.70 per day (29-252 per year). The labile fraction was characterized by a decay constant of ~0.02 per day (~7 per year), with a range of 0.002-0.053 per day (7.2 and 0.72-19 per year). The very labile and labile fraction constituted 32% and 50% of total DOP, respectively. RetroMOPS presented here focuses on the dominant labile fraction; its maximum possible rate for DOP decay for optimization is 7.2 per year, the observed average decay rate of the labile DOP in Hopkinson et al. (2002). Note that, however, most of the simulated ocean is far off the productive shelf areas; further, DOP in the model mimicks a variety of biogeochemical components (possibly even bacteria, or other non-sinking dead organic particles), and thus the observations may not be directly transferable to simulated DOP. In a three-step optimization process Letscher et al. (2015a) optimized a global model of semi-labile and refractory DOM against observations, and found rates of 0.016 per year for semilabile DOP at the surface, and 0.22 per year for semilabile DOP in the mesopelagical. Production and turnover rates for refractory DOP were very small, except for an additional photo-oxidation rate of 0.07 per year. The optimum decay rate of 0.47 per year found in this study is within the range estimated by Letscher2015; also, the nonrequirement of fast surface turnover agrees with their results, which point towards lower remineralization of DOP at the sea surface. I will add more details on the range of potential decay rates, and my particular choice for boundaries in a revised version of the paper.

==Acknowledgements==

"Pg. 13: Is this paper part of a special issue or wider celebration of the life of Ernst Maier-Reimer?; if so, an earlier note in the introduction would seem to be in order; if not, it may be worth making the rationale for this tribute a little clearer (e.g. note Maier-Reimer's recent passing)"

IK: It is part of a special issue in memory of Ernst Maier-Reimer, and there is an introduction by Christoph Heinze. I here just wanted to add my personal acknowledgment.

==Figures and Tables==

Table 1: Why is the ostensibly fixed parameter DINmin very slightly different in the two RetroMOPS runs?

IK: This was a typo, I correct this to 15.80 for RetroMOPSr

Reply to referee 4:

I thank referee 4 for his/her critical yet helpful comments. Below is my reply, indicated by "IK"  $\,$

"My overall feeling is that the paper is badly lacking in focus. Reading through I was always struggling to understand what major point the author was hoping to make. Is it that the simple model is nearly as good as the complex model, or is it that different parts of the model are better constrained by different kinds of observations? At the moment the article reads as if two separate (and somewhat poorly developed) stories have been combined into one, with very little thought as to what connects them. I think that the author either needs to pick one theme, and develop it better, or needs to do a much better job of finding a narrative thread linking the two themes together. It is up to the author to identify how that might be achieved."

IK: I am sorry that the paper appears to be so unfocused, and will try to explain my reasoning better here, and in a revised version of the paper. In short, given the sometimes high structural complexity of global biogeochemical models there are only sparse observations to constrain them, the two main findings of the paper:

- complex, biological dynamics are not well constrained by a rather biogeochemical misfit to nutrients and oxygen

- the simple model performs almost as good as the more complex one, with respect to the given misfit function are somehow connected. Calibrating a complex model would possibly require either a much more complex misfit function (with respect to observations; e.g. using Chl a, observations of zooplankton abundance, and DOP). Given that

- models of higher complexity, such as MOPS, are usually applied to research questions that relate to more biogeochemical issues (such as ocean carbon inventory, or deoxygenation)

- these models are expensive in terms of computing time, thereby hampering exploitation of model (parameter) sensitivity and skill in spun up state, and

- more "sophisticated" data sets are sparse, and many of the observations may contain a high uncertainty, or noise

I find it important to raise some awareness about the necessary level of model complexity, and the uncertainty associated with model structure and parameters. In some cases it may be more approriate to spend more time on carefully exploiting the parameter space instead of adding more complexity. Of course, this tightly relates to the research question addressed with the model. I will do my best to render the paper more focused and clear in a revised version.

Specific comments

Abstract

"Line 7: "a complex seven-component model (MOPS), and a very simple twocomponent model (RetroMOPS)" and "The simple model, which contains only nutrients and dissolved organic phosphorus (DOP)". RetroMOPS clearly has four components: PO4, NO3, O2 and POM."

IK: Yes, thank you. This will be corrected.

"Line 13: Please do a better job of explaining what is "the global bias"."

IK: I will add "(global inventory of oxygen and fixed nitrogen)"

1 Introduction

"Line 29: "[Kriest et al. (2017)] showed that annual mean tracer concentrations do not provide much information on parameters related to the dynamic biological processes taking place in the euphotic zone". Should be "annual mean tracer concentrations did not provide much information", as I am not convinced this is a general result for all models."

IK: It would be interesting to see other models when facing optimisation against the same misfit (volume weighted RMSE of annual mean nutrients and oxygen); Until then, I agree, it should be "did".

**2.2.1 Primary production**

"Equation 1: Why use the mean phytoplankton concentration at all? It would be more consistent with the rest of the model (i.e. Equation 5) to convolve the specific growth rate and the phytoplankton concentration into a single growth rate of the phytoplankton population (mmol P m-3 d-1)."

IK: I used this particular decomposition to clearly illustrate the specific assumptions that may be inherent in simple models such as RetroMOPS (similar to Kriest et al., 2010). In addition, deriving the specific growth rate from MOPS, and transferring this to RetroMOPS, would involve accounting for nutrient concentration and limitation – which in turn depend on the remineralisation rate and sinking speed. I therefore chose this way of aligning both models, and would prefer to keep it that way. Note that the resulting specific growth rates (including limitation by nutrients, temperature and light) between both models are not too different: 0.1021 d-1 (RetroMOPS) and 0.1267 d-1 (MOPS). I will add a sentence on this in the revision.

 $\ensuremath{\texttt{2.2.2}}$  The fate of primary production: Export, DOP production and remineralisation

"Line 19: "DOP then decays to phosphate and nitrate". To me it would make sense to call it POM."

IK: POM would be something that sinks, which clearly distinguishes it from DOM.

"Line 19: "To allow for a potential, fast recycling loop at the surface, RetroMOPS parameterises an additional decay rate". Presumably this is inspired by (Oschlies 2001), but why would this be necessary in the absence of assimilated primary production observations?"

IK: There are three reasons why I have embedded this fast recycling loop: first, DOP production and decay in RetroMOPS has to mimick all dynamic surface processes of MOPS, so I initially expected it to require a specific degradation rate constant for the surface. As it turned out, this is not necessary (this parameter during optimisation was reduced to nearly zero). Second, at a later stage it might indeed be interesting (and helpful) to include primary production into the misfit function, with possibly different resulting best parameters surface DOP decay. Finally, data by Hopkinson et al (2002) indicate that DOP recycling rates may be much higher than commonly applied in global models. I will add some discussion on this in a revised version of the paper.

"Equation 4: I think a bit more could be said about the interdependence of sO2(j)and sDIN(j). For example, their sum forms a coefficient for remineralisation, so it is important to note that their sum is constrained between 0 and 1."

IK: I will comment more on this function in a revised version of the paper.

2.5 Misfit function

"Equation 11: I am a bit confused by how the misfit function and its components are defined. In particular, I cannot see how o  $\hat{I}D^{\sim}$  (the global average observed concentration of the respective tracer) is included in the RHS. "

IK: This was a mistake by me; The RHS was missing  $1/bar(o_j)$  after the first sum, but it should have been after J(j). Thank you for drawing my attention to this.

"Also, it seems that the model is being compared to gridded observations, instead of observational equivalents being extracted at the spatiotemporal locations of the observations. As the gridding process will introduce its own set of errors, this choice needs some justification."

IK: Although regridding the observations onto the coarser model grid removes much of the variability in the observations, this procedure is much more efficient (in terms of computing time) during the optimisation process. Further, by following this approach the model is not penalised for its apparent lack of resolution. It could be worthwhile adding the variance, that arises from the regridding process, and the variance in the data themselves, as weight to the misfit function. However, in an earlier study (Kriest et al., 2010) we could not find any large effects of this on model assessment. Testing different misfit function with respect to observational data sets, weighting schemes, etc., will be subject of follow-up work, but possibly exceed the scope of this paper.

2.6 Optimisation of MOPS

"Line 15: I don't think including results from the hand-tuned model brings anything of value to the paper."

IK: I disagree: as "hand-tuning" still seems to be common practice in global biogeochemical modelling, I think it merits some presentation and discussion.

3.1 Optimisation of MOPS

"Line 4: Fig 1 (rather than Fig S1)?"

IK: Both - this will be changed to "Figures 1 and S1".

"Line4:10%and1%(ratherthan10%and1âU°e/âU°e/âU°e/)?"

IK: Yes, thank you for noting this!

"Line 12 (and throughout the text): "reduced denitrification". It is probably safer to avoid the word reduced except with regard to the chemical process."

IK: I agree, and will exchange "reduced" by "lower".

3.3 Optimisation of RetroMOPS

"Line 21: "The misfit to phosphate (Fig. 8, lower left panel) shows an elongated valley in the two-dimensional projection". I do not see a valley in this Figure. The misfit slopes down from the top-right towards the bottom left, but there is nothing to suggest it slopes back up again after reaching a minimum."

IK: I was referring to the - admittedly few - green and red points in the lower right corner, that indicate an increase in the misfit function. I will replace "shows" by "indicates".

[revised manuscript text omitted]

$$S^{\text{PO4}}(\underline{k}) = -P(\underline{k}) - P_{\text{production}} + \underline{\lambda_{\text{sDOP}} \text{ DOP} } \underline{\lambda_{\text{s}} \text{ DOP} \text{ surface decay}} + \underline{D}(\underline{k}) + \underline{\lambda_{\text{DOP}} \text{ DOP}(\underline{k})} \underline{s_{\text{O2}}(\underline{k}) + s_{\text{DIN}}(\underline{k})} \underline{[D + \lambda \text{ DOP}]} \underline{[s_{\text{O2}}]} \\ \text{15} \quad S^{\text{DOP}}(\underline{k}) = \sigma_{DOP} P(\underline{k}) \underline{\sigma_{DOP}} P_{\text{release}} - \underline{\lambda_{\text{sDOP}} \text{ DOP} } \underline{\lambda_{\text{s}} \text{ DOP} \text{ surface decay}} - \underline{\lambda_{\text{DOP}} \text{ DOP}(\underline{k}) s_{\text{O2}}(\underline{k}) + s_{\text{DIN}}(\underline{k})} \underline{\lambda_{\text{DOP}} \text{ [s_{\text{O2}} + s_{\text{DIN}}]}} \\ S^{\text{O2}}(\underline{k}) = R_{-O2:P} P(\underline{k}) R_{-O2:P} P_{\text{production}} - R_{-O2:P} \underline{\lambda_{\text{sDOP}} \text{ DOP} R_{-O2:P} } \underline{\lambda_{\text{sDOP}} \text{ DOP} s_{\text{surface decay}}} - R_{-O2:P} D(\underline{k}) + \underline{\lambda_{\text{DOP}} \text{ DOP}^{*} s_{\text{O2}}} \\ S^{\text{DIN}}(\underline{k}) = -dP(\underline{k}) - dP_{\text{production}} + S^{NFix}_{DIN}(\underline{k}) S_{N-\text{fixation}} + d\underline{\lambda_{\text{sDOP}} \text{ DOP} d\lambda_{\text{s}} \text{ DOP} s_{\text{surface decay}}} + \underline{D}(\underline{k}) + \underline{\lambda_{\text{DOP}} \text{ DOP} s_{\text{sO2}}(\underline{k})} \\ \end{array}$$

[revised manuscript text omitted]
 RetroMOPSo targets at these parameters, namely  $\sigma_{\text{DOP}}$ ,  $\lambda_{\text{sDOP}}$  and  $\lambda_{\text{DOP}}$ .

- While  $\sigma_{\text{DOP}}\sigma$ ,  $\lambda_s$  and  $\lambda$ . While  $\sigma$ , as parameter that regulates the export ratio, may be more or less well constrained, 15  $\lambda_{\text{sDOP}}$  and  $\lambda_{\text{DOP}}$  and  $\lambda$  both include a variety of processes, which may act on time scales of days to years. In a set of nine a priori sensitivity experiments the effect of these parameters on the misfit function is explored by varying  $\lambda_{\text{DOP}}$ Hopkinson et al. (2002) applied a multi-G model to incubations of DOP sampled in surface waters of the middle Atlantic Bight, and measured decay constants for the very labile fraction (32% of total DOP) of  $\approx 80 \text{ y}^{-1}$ , with a range of 3-254 y-1. Half of total DOP was in the labile fraction and characterised by a decay constant of  $\approx 7 \text{ y}^{-1}$ , ranging from 0.8-43 y-1. However,
- these observations may not be directly transferable to globally simulated DOP, because most of the simulated ocean is far off the productive shelf areas; further, DOP in RetroMOPS is assumed to mimic a variety of biogeochemical components and processes. In a three-step optimisation study Letscher et al. (2015), who optimised a global model of semi-labile and refractory DOM against observations estimated rates of 0.016 y-1 for semilabile DOP at the surface, and 0.22 y-1 for semilabile DOP in the mesopelagial, i.e. much lower than suggested by Hopkinson et al. (2002). Summarising, the potential decay rate of the very labile to semi-labile fraction varies over several orders of magnitude, from O(0.01) O(100) y-1.
- Optimisation of RetroMOPS focuses on the dominant labile to semi-labile fraction, but allows for some potential fast turnover rates of DOP at the sea surface (towards the values observed by Hopkinson et al., 2002). To obtain a first impression on model sensitivity towards these parameters, a set of nine a priori experiments, that vary  $\lambda$  between 0.18 y-1 and 0.72 y-1 , and  $\lambda_{sDOP}$  and  $\lambda_s$  between 0 y-1 and 0.36 y-1 (see tablehas been carried out (Table 2). The results of these sensitivity

[revised manuscript text omitted]
                            | $\mathrm{MOPS}^{\mathrm{r}}$ | $\mathrm{MOPS}^{\mathrm{oS}}$ | $\mathrm{MOPS}^{\mathrm{oD}}$ | $RetroMOPS^{\rm r}$    | $RetroMOPS^{\circ}$ | unit                             |
|---------------------------------------|------------------------------|-------------------------------|-------------------------------|------------------------|---------------------|----------------------------------|
| $\sigma_{\text{DOP}} \sigma_{\sim}$   | -                            | -                             | -                             | 0.67                   | [0.4 - 0.8]         |                                  |
| $\lambda_{sDOP} \lambda_{s}$          | -                            | -                             | -                             | 0                      | [0.0 - 3.6]         | $y^{-1}$                         |
| $\lambda_{\text{DOP}} \lambda_{\sim}$ | 0.17                         | 0.17                          | 0.17                          | 0.36                   | [0.036 - 3.6]       | $y^{-1}$                         |
| $I_{\rm c}$                           | 24                           | [4 - 48]                      | 9.65                          | 9.65                   | 9.65                | ${ m W}{ m m}^{-2}$              |
| $K_{\rm PHY}$                         | 0.03125                      | [0.001 - 0.5]                 | 0.5                           | 0.5                    | 0.5                 | mmol P m $^{-3}$                 |
| $\mu_{\rm ZOO}$                       | 2                            | [1 - 3]                       | 1.89                          | -                      | -                   | $d^{-1}$                         |
| $\kappa_{\rm ZOO}$                    | 3.2                          | [1.6 - 4.8]                   | 4.55                          | -                      | -                   | $(d \text{ mmol P m}^{-3})^{-1}$ |
| $b^*$                                 | 0.858                        | [0.4 - 1.8]                   | [0.4 - 1.8]                   | 1.0725                 | [0.4 - 1.8]         |                                  |
| $R_{-O2:P}$                           | 170                          | [150 - 200]                   | [150 - 200]                   | 171.7                  | 171.7               | mmol O 2 :mmol P      |
| $\mu_{ m NFix}$                       | 2                            | 2                             | [1 - 3]                       | 1.19                   | 1.19                | nmol N $d^{-1}$                  |
| $DIN_{\min}$                          | 4                            | 4                             | [1 - 16]                      | <del>15.79</del> 15.80 | 15.80               | mmol N m $^{-3}$                 |
| $K_{O2}$                              | 2                            | 2                             | [1 - 16]                      | 1.0                    | 1.0                 | mmol $O_2 m^{-3}$                |
| $K_{\rm DIN}$                         | 8                            | 8                             | [2 - 32]                      | 31.97                  | 31.97               | mmol N m $^{-3}$                 |

\* Note that from b (the optimised parameter) in MOPS we calculate the rate of vertical increase in sinking speed a of w = a z, via a = r/b. For r we assume nominal detrital remineralisation of  $r = 0.05 \text{ d}^{-1}$ . The resulting values for a are: 0.058275 (b = 0.858), 0.0278 (lower boundary) and 0.125 (upper boundary).

**Table 2.** Results (misfit *J*) of sensitivity experiments with model RetroMOPS, regarding parameters  $\frac{\lambda_{\text{sDOP}}}{\lambda_{\text{s}}}$  and  $\frac{\lambda_{\text{DOP}}}{\lambda_{\text{s}}}$  for DOP decay rate. The misfit of the reference scenario RetroMOPSr is indicated in bold.

|                                                    | $\frac{\lambda_{\rm sDOP} = 0}{\lambda_{\rm s}} = 0$ | $\lambda_{\text{sDOP}} = 0.18 \lambda_{\text{s}} = 0.18$ | $\lambda_{\text{sDOP}} = 0.36 \lambda_{\text{s}} = 0.36$ |
|----------------------------------------------------|------------------------------------------------------|----------------------------------------------------------|----------------------------------------------------------|
| $\lambda_{\text{DOP}} = 0.18 \cdot \lambda = 0.18$ | 0.502                                                | 0.480                                                    | 0.480                                                    |
| $\lambda_{\rm DOP} = 0.36 \ \lambda = 0.36$        | 0.466                                                | 0.476                                                    | 0.493                                                    |
| $\lambda_{\rm DOP} = 0.72 \cdot \lambda = 0.72$    | 0.503                                                | 0.522                                                    | 0.539                                                    |

**Table 3.** Optimisation results: minimum misfit  $J^*$ , optimum parameters and their uncertainties. To determine parameter uncertainty, we selected a group  $\Omega$  of the 1% best individuals, i.e. individuals defined by a misfit  $J_i : J_i/J^* - 1 \le \Delta_J$ , with  $\Delta_J = 0.001$ . The number of these individuals  $N(\Omega)$  is also denoted as fraction  $n(\Omega)$  of all individuals of the optimisation  $\lambda \times N$ , where N is the number of generations, and  $\lambda = 10$  the population size. For each parameter  $\Theta$  the first column gives the optimal parameter  $\Theta^*$  (i.e., the average parameter of the last generation). The second and third column present the parameter range of all individuals of  $\Omega$ , expressed as absolute value  $(R_{\Theta}(\Omega))$ , and normalised by the a priori range of parameters  $(R_{\Theta}^{A}; \text{ see Table 1}): r_{\Theta}(\Omega) = R_{\Theta}(\Omega)/R_{\Theta}^{A}$  value.

| Experiment:                          |            | $\mathrm{MOPS}^{\mathrm{oS}}$ |                      |            | $MOPS^{oD}$ $MOPS^{oD}_*$ |                      |            | RetroMOPS o |                      |            |                      |                      |
|--------------------------------------|------------|-------------------------------|----------------------|------------|---------------------------|----------------------|------------|------------------------|----------------------|------------|----------------------|----------------------|
| Parameter                            | $\Theta^*$ | $R_{\Theta}(\Omega)$          | $r_{\Theta}(\Omega)$ | $\Theta^*$ | $R_{\Theta}(\Omega)$      | $r_{\Theta}(\Omega)$ | $\Theta^*$ | $R_{\Theta}(\Omega)$   | $r_{\Theta}(\Omega)$ | $\Theta^*$ | $R_{\Theta}(\Omega)$ | $r_{\Theta}(\Omega)$ |
| <del>σDOP σ</del>         | -          | -                             | -                    | -          | -                         | -                    | -          | -                      | -                    | 0.73       | [0.7-0.7]            | 6                    |
| $\lambda_{\rm sDOP} \lambda_{\rm s}$ | -          | -                             | -                    | -          | -                         | -                    | -          | -                      | -                    | 0.02       | [-0.1-0.2]           | 8                    |
| $\lambda_{\text{DOP}}\lambda_{\sim}$ | -          | -                             | -                    | -          | -                         | -                    | -          | -                      | -                    | 0.47       | [0.4-0.5]            | 4                    |
| $I_{\rm c}$                          | 9.66       | [8.9-10.3]                    | 3                    |            |                           |                      |            |                        |                      |            |                      |                      |
| $K_{\rm PHY}$                        | 0.50       | [0.4-0.5]                     | 28                   |            |                           |                      |            |                        |                      |            |                      |                      |
| $\mu_{\rm ZOO}$                      | 1.89       | [1.6-2.0]                     | 22                   |            |                           |                      |            |                        |                      | -          | -                    | -                    |
| $\kappa_{\rm ZOO}$                   | 4.57       | [3.0-4.7]                     | 53                   |            |                           |                      |            |                        |                      | -          | -                    | -                    |
| $b^{\S}$                             | 1.34       | [1.3-1.4]                     | 4                    | 1.39       | [1.4-1.4]                 | 3                    | 1.41       | [1.4-1.4]              | 2                    | 0.98       | [1.0-1.0]            | 2                    |
| $R_{-O2:P}$                          | 167.0      | [165-170]                     | 9                    | 171.7      | [170-173]                 | 6                    | 174.9      | [174-176]              | 5                    |            |                      |                      |
| $\mu_{\rm NFix}$                     |            |                               |                      | 1.19       | [1.1-1.4]                 | 13                   | 1.47       | [1.4-1.6]              | 10                   |            |                      |                      |
| $DIN_{\min}$                         |            |                               |                      | 15.80      | [13-16]                   | 20                   | 12.96      | [12-16]                | 25                   |            |                      |                      |
| $K_{O2}$                             |            |                               |                      | 1.00       | [0.3-1.8]                 | 10                   | 1.00       | [0.5-1.4]              | 6                    |            |                      |                      |
| $K_{\rm DIN}$                        |            |                               |                      | 31.97      | [30-34]                   | 12                   | 31.97      | [22-33]                | 35                   |            |                      |                      |
| $J^*$                                |            | 0.450                         |                      |            | 0.439                     |                      |            | 0.427                  |                      |            | 0.458                |                      |
| $\lambda \times N$                   |            | 1820                          |                      |            | 1190                      |                      |            | 2000                   |                      |            | 660                  |                      |
| $N(\Omega)$                          |            | 718                           |                      |            | 514                       |                      |            | 1285                   |                      |            | 262                  |                      |
| $n(\Omega)$                          |            | 39                            |                      |            | 43                        |                      |            | 64                     |                      |            | 40                   |                      |

**Table 4.** Global annual fluxes of primary production (P), grazing (GRAZ), fixed nitrogen loss through pelagic denitrification (NLOSS), export production (F120, flux through 120 m), flux through 2250 m (F2250), and benthic burial (BUR), in Pg N  $y^{-1}$ , for the reference experiment of MOPSr, MOPSoD, MOPSoD, MOPSoD and RetroMOPS, for which we show the fluxes of the (best) reference experiment, RetroMOPSr, the range of all sensitivity experiments, and the optimised run, RetroMOPSo. Also shown are some globally derived, observed estimates. Conversion between different elements was carried out via N:P=16, and C:P=122.

| Experiment                      | Р         | GRAZ      | NLOSS       | F120        | F2250       | BUR         |
|---------------------------------|-----------|-----------|-------------|-------------|-------------|-------------|
| MOPS r               | 5.44      | 3.52      | 0.098       | 0.918       | 0.107       | 0.051       |
| $\mathrm{MOPS}^{\mathrm{oS}}$   | 7.52      | 4.74      | 0.117       | 1.102       | 0.056       | 0.018       |
| $\mathrm{MOPS}^{\mathrm{oD}}$   | 7.70      | 4.97      | 0.068       | 1.080       | 0.055       | 0.022       |
| $\mathrm{MOPS}^{\mathrm{oD}}_*$ | 7.80      | 5.06      | 0.083       | 1.081       | 0.053       | 0.021       |
| RetroMOPS r          | 5.56      | -         | 0.078       | 1.194       | 0.043       | 0.010       |
| RetroMOPS (range)               | 4.88-6.21 | -         | 0.076-0.084 | 1.076-1.286 | 0.039-0.047 | 0.008-0.014 |
| RetroMOPS o          | 6.31      | -         | 0.071       | 1.12        | 0.052       | 0.009       |
| Observed §           | 7.68-8.09 | 4.79-5.71 | 0.05-0.08   | 0.29-1.53   | 0.03-0.07   | 0.02        |

§ Observed fluxes are from Carr et al. (2006, primary production), Honjo et al. (2008, particle flux), Lutz et al. (2007, particle flux), Dunne et al. (2007, particle flux), Schmoker et al. (2013, primary production, zooplankton grazing excluding/including mesozooplankton grazing), Wallmann (2010, burial; without shelf and slope region), and Kriest and Oschlies (2015, fixed nitrogen loss).

---

## Author Response (AR2)

I thank the referees for their careful reading and helpful comments. My responses below to this second round of review are marked with IK2.

Report 1 (reviewer 1):

The clarity of the manuscript and the depth of discussions have improved in this revision. Although I agree that this study presents a more objective assessment of the models with different structures compared with the Kriest et al. 2010 paper, the novelty of this study is still weak. Nevertheless, I believe that this study can serve as a step towards more thorough and systematic assessments of ocean biogeochemistry models with different complexities.

There are some minor points that need to be clarified further.

1. The two models presented here have different complexities in terms of the number of variables (or components): 7 variables for MOPS and 4 variables for RetroMOPS. However, both models have the same number of tunable parameters, i.e., total 11 parameters. Given the same number of tunable parameters in both models, I am not sure if we can call them as models with different complexities, e.g., "complex" and "simple" as indicated in title and throughout the text.

IK2: Beside the parameters listed in Table 3, MOPS has 9 more tunable parameters (8, if I subtract the surface DOP decay rate of RetroMOPS) than RetroMOPS, which related to phytoplankton mortality and exudation, zooplankton half-saturation constant for food, its grazing efficiency, linear mortality rates and detritus remineralization rate. I include one sentence on this at the end of subsection 2.2.5, to point this out.

2. In lines #10-11 of abstract, "The simple model is sensitive to the parameterization of DOP production and decay". Does the author mean that the simple model's misfit is sensitive, or that the simulated tracer distributions from the simple model are sensitive, or something else? Please be more precise.

IK2: I replace this with "The simple model contains only nutrients, oxygen and dissolved organic phosphorus (DOP). Its misfit and large scale tracer distributions are sensitive to the parameterisation of DOP production and decay."

3. In line #15, Does the author mean phosphorus? - IK2: Yes, corrected.

4. In line #29, "parameters related to remineralization". In fact, all of the parameters including b are related to remineralization". Please be more precise. - IK2: I have exchanged this by "related to oxidant-affinity of remineralization".

5. In line #31, double "range" - IK2: Corrected.

Report 2 (reviewer 4):

The author has made a good effort to incorporate most of my suggestions. Where suggested corrections are not made, justification was provided. I therefore recommend to accept the paper, subject to some very minor corrections.

Abstract, line 5: "Examination *of* the models' fit to climatologies of inorganic tracers" - IK2: Corrected.

Abstract, line 10: "The simple model, which contains only *two* nutrients, oxygen and dissolved organic phosphorus" - IK2: Corrected.

Page 3, line 11: "The latter procedure may help to elucidate which level of complexity is required" (remove comma)  - IK2: Corrected.

Page 3, line 15: "parameters related *to* oxidant-affinity of remineralisation"  - IK2: Corrected.

Page 3, line 24: "simulates fully prognostic changes in surface production" - IK2: Corrected.

Page 6, line 1: "resulting in sO2(j)+sDIN(j) = 1". Should this be equal to 1, or between 0 and 1? If the former (as written), equation 4 is superfluous. – IK2: no, these will indeed add up to 1.

Page 11, line 3: "The good determination of b by dissolved inorganic tracers is *in* agreement with earlier studies" – IK2: Corrected.

Page 15, line 32: "Results suggest that little evidence that higher biological complexity implies better model performance in reproducing observed global- scale bulk properties." (Kwiatkowski et al., 2014). Quotations should be verbatim. Paraphrased text should not be placed in quotes. – IK2: I am sorry, a "that" slipped into the quotation, and I missed "of ocean biogeochemistry"; I have corrected that.

Page 16, line 24: "parameters relevant for large-scale, global distributions of oxygen ... could be determined with a high fidelity". Precision would probably be a better word than fidelity (high fidelity to what?) – IK2: I have exchanged "with a high fidelity" by "well".

Report 3 (reviewer 4):

My thanks to the author for responding at length to my comments. In general, I am happy with the responses to my original comments. I have a few remaining comments, but these are also relatively minor, and it is even possible that they have been addressed already. I also have a "major" comment — see immediately below this paragraph — but I think this is more for the journal than for the author. In general, my recommendation is publish following minor revision.

Per my remark above, my "major" comment is that it is not easy to properly understand how the author has modified their manuscript in response to referee comments. The author responses are very much in the form of "I will alter the text to address this" statements rather than "I have altered the text as follows to address this" statements. As a result, I cannot be entirely sure about how the manuscript has changed. To be fair, the author has included a tracked changes version of the manuscript but it is not a straightforward task to work out what exactly has changed in response to particular criticisms. I suspect that this stems from the journal's preference for authors to respond to comments in a sort of dialogue with their referees before revising the manuscript, rather than the more conventional "response + revised manuscript" mode. In any case, I have generally taken the author at face value on the content of their changes in response to my criticisms, but have focused where changes are less clear / more contentious.

In the text below, I leave the "IK" designations in place to denote author comments. My new comments are marked up as "REF"

IK: This comment by the reviewer addresses two points: (a) the (lack of) introduction of RetroMOPS, as well as (b) its performance with respect to the skill metrics to biogeochemical tracers.

REF: I think I have misled the author as to what I was suggesting here. By drawing attention to the Kwiatkowski et al. (2014) paper, I was merely making the passing observation that increasing complexity does not necessarily lead to increasing performance. I certainly didn't intend to suggest that a particular intercomparison was needed, nor that the authors of that paper presented anything like a definitive guide as to how this should be done (its choice of metrics is more a "lowest common denominator" analysis). As such, section 3.3 of the revised manuscript, while largely welcome, may over-emphasise the latter.

IK2: I agree, and have now skipped everything from p. 15, line 35 "This conclusion [...]" to p. 16, line 5.

IK: In the model nitrogen fixation balances denitrification on large time and space scales. It depends on biogeochemical parameters and circulation, how fast the two processes are connected (see also Kriest and Oschlies, 2015). I will add "in the long term" in a revised version of the manuscript, and add some words on the potential spatial

distinction.

REF: This is fine, but there's an earlier mention of nitrogen fixation for MOPS that makes the comment below but does not qualify the time/space scales point:

"Loss of fixed nitrogen is balanced by a simple parameterisation of nitrogen fixation by cyanobacteria, which relaxes the nitrate-to-phosphate ratio to d with a time constant, $\mu^{\shortleftarrow}$NFix."

IK2: I now mention the time and space scales twice: Once when briefly presenting MOPS, after the above sentence: "In the long term nitrogen fixation balances the simulated loss of fixed nitrogen via denitrification, although they may occur in distant areas (see Kriest and Oschlies, 2015, for more details)." When presenting RetroMOPS, I write "As in MOPS, on long time scales nitrogen fixation balances the simulated loss of fixed nitrogen via denitrification, although the regions of nitrogen loss and gain can be spatially segregated (Kriest and Oschlies, 2015)." in section 2.2.4

REF: Just to clarify, but when I previously said "Pg. 7, lines 24-25: While the normalisation to global concentrationshould help with N and P (since they are related linearly through the ocean)", what I meant was that N and P in the real ocean have a relatively strong linear relationship (unlike, say, either element and silicon), not that they are (or should be) tightly linearly controlled in the model. Sorry for any confusion on this point.

IK: In MOPS' optimisations, $R\_-O2:P$ did not show any significant deviations from its default value of 170, so I decided to keep this fixed, and only vary those parameters, that relate to the parts of RetroMOPS that are very different to MOPS (see above). I will try to explain this better in the revision.

REF: This line of argument sounds sensible. However, I can't find it being mentioned explicitly (or clearly so) in the manuscript. While I can see that the value of RetroMOPS is that of the optimised MOPS, it isn't clear enough to me that this has been deliberately done. Also, this approach also overlooks the faint possibility that while a parameter is insensitive in one model, that it might not be in another. An acknowledgement of this might be worth making.

IK2: I have added a half-sentence stating this in the last paragraph of section 2.7.

IK: Because the reduced data set did not show any large changes in estimated parameters or misfit of MOPS, and because I assume that other issues might be more important to investigate in an optimisation context (circulation; components and form of the misfit function), I decided not to spend computational resources on RetroMOPS with reduced data set. This "reduced data" experiment was mainly aimed at investigating the potential effect of circulation in the equatorial Pacific on the parameter estimate. Possibly due to its small area, the effect is almost negligible.

REF: Is this acknowledged in the text at all?

IK2: No, not really, thank you for mentioning it. I now added in section 3.1, fourth paragraph "The only moderate effect of the eastern equatorial Pacific on optimization is likely related to the small volume occupied by this region, compared to total ocean volume." and also "likely because of the small volume of this region" at the end of the second paragraph of the conclusion section, and also

REF: Pg. 16, line 19: correct "mesopelagial" to "mesopelagic".

IK2: Corrected.

[revised manuscript text omitted]

Supplement:

[Figure]

**Figure S1.** Diagram illustrating the changes for the downscaling of model MOPS (gray) to RetroMOPS. Omitted compartments are indicated by red diagonal bars. Structural changes for fluxes between the compartments are indicated by red borders.

[Figure]

**Figure S2.** Model misfit of the best (final) candidate of MOPS$^{oD}$ (red) and RetroMOPS$^{o}$ (blue), plotted over the entire spin-up of 3000 years (as log scale). Left panel: total misfit $J$, divided by three. Left to right panels: components (phosphate, nitrate, oxygen) of the misfit function. Note that, depending on simulation time, a different model type may exhibit the lowest misfit.

[Figure]

**Figure S3.** Model misfit, plotted for each pair of parameter combinations of MOPS$^{oD}$. Colour indicates misfit (see the colour bars on the right). A circle indicates the parameter of one individual of the last generation. For better visibility the parameter range to its boundaries (see Table 1).

[Figure]

**Figure S4.** Parameter distribution of model simulations obtained during the optimisation of $\text{MOPS}^{\text{oD}}_{*}$, whose misfit do not exceed a threshold limit of $\Delta J = 1.1\,J^{*}$ (10%, red bars) or $\Delta J = 1.01\,J^{*}$ (1%, open bars) of the minimum misfit $J^{*}$. For the projection parameters of all model simulations in the optimisation trajectory were grouped into 50 classes.

[Figure]

**Figure S5.** Zonal mean oxygen for three different basins of two sensitivity experiments with RetroMOPS and observations. Upper panels: RetroMOPS with low DOP recycling ($\lambda_s = 0$, $\lambda = 0.18$). Lower panels: RetroMOPS with high DOP recycling ($\lambda_s = 0.36$, $\lambda = 0.72$). Mid panels: observations. Contour lines show simulated zonal average DOP (dashed: 0.1 mmol P m$^{-3}$. thick: 0.5 mmol P m$^{-3}$).

[Figure]

**Figure S6.** Zonal mean N* (NO₃-PO₄ × 16 for three different basins of two sensitivity experiments with RetroMOPS and observations. Upper panels: RetroMOPS with low DOP recycling ($\lambda_s = 0$, $\lambda = 0.18$). Lower panels: RetroMOPS with high DOP recycling ($\lambda_s = 0.36$, $\lambda = 0.72$). Mid panels: observations. Contour lines show simulated zonal average annual fixed nitrogen loss through denitrification (thin: 1 $\mu$mol N m⁻³y⁻¹, dashed: 10 $\mu$mol N m⁻³y⁻¹ thick: 100 $\mu$mol N m⁻³y⁻¹).

[Figure]

**Figure S7.** Model misfit, plotted for each pair of parameter combinations of RetroMOPS°. Colour indicates misfit (see the colour bars on the right). A circle indicates the parameter of one individual of the last generation. For better visibility the parameter range to its boundaries (see Table 1).